# Dimensionality Reduction for Wasserstein Barycenter

**Zachary Izzo**
Stanford University
zle.izzo@gmail.com

**Sandeep Silwal**
MIT
silwal@mit.edu

**Samson Zhou**
Carnegie Mellon University
samsonzhou@gmail.com

## Abstract

The Wasserstein barycenter is a geometric construct which captures the notion of centrality among probability distributions, and which has found many applications in machine learning. However, most algorithms for finding even an approximate barycenter suffer an exponential dependence on the dimension $d$ of the underlying space of the distributions. In order to cope with this "curse of dimensionality," we study dimensionality reduction techniques for the Wasserstein barycenter problem. When the barycenter is restricted to support of size $n$, we show that randomized dimensionality reduction can be used to map the problem to a space of dimension $O(\log n)$ independent of both $d$ and $k$, and that *any* solution found in the reduced dimension will have its cost preserved up to arbitrary small error in the original space. We provide matching upper and lower bounds on the size of the reduced dimension, showing that our methods are optimal up to constant factors. We also provide a coreset construction for the Wasserstein barycenter problem that significantly decreases the number of input distributions. The coresets can be used in conjunction with random projections and thus further improve computation time. Lastly, our experimental results validate the speedup provided by dimensionality reduction while maintaining solution quality.

## 1 Introduction

The Wasserstein barycenter (WB) is a popular method in statistics and machine learning for summarizing data from multiple sources while capturing their underlying geometry [AC11a]. The problem is defined as follows. Suppose we have a collection of data, represented as $k$ discrete probability distributions $\mu_1, \ldots, \mu_k$ on $\mathbb{R}^d$. Given a set of non-negative weights $\lambda_1, \ldots, \lambda_k$ that sum to 1, and a class $\mathbb{P}$ of probability distributions on $\mathbb{R}^d$, a Wasserstein barycenter under the $L_p$ objective for a parameter $p > 0$ is a probability distribution $\nu \in \mathbb{P}$ that minimizes

$$\sum_{i=1}^{k} \lambda_i W_p(\mu_i, \nu)^p, \tag{1}$$

where $W_p(\mu_i, \nu)$ is the $p$-Wasserstein distance.

The Wasserstein barycenter is a natural quantity that captures the geometric notion of centrality among point clouds, as it utilizes the optimal transport distance [BT97] between a number of observed sets. Thus, Wasserstein barycenters have been extensively used in machine learning [SLD18], data sciences [RU02, EHJK20], image processing [RGT97], computer graphics [PW09], and statistics [Vil08], with applications in constrained clustering [CD14, HNY$^+$17], Bayesian learning [SLD18], texture mixing [RPDB11], and shape interpolation [SDGP$^+$15].

Unfortunately, the problem is NP-hard to compute [AB21, BP21] and many algorithms that even approximate the Wasserstein barycenter suffer from large running times, especially if the datasets are high dimensional [MC19]. Indeed, [ABA21] recently gave an algorithm that computes the

35th Conference on Neural Information Processing Systems (NeurIPS 2021).

Wasserstein barycenter using runtime that depends exponentially on the dimension, thus suffering the "curse of dimensionality."

To alleviate these computational constraints, we consider dimensionality reduction for computing the Wasserstein barycenter. Dimensionality reduction can be used to improve the performance of downstream algorithms on high dimensional datasets in many settings of interest, e.g., see the survey [CG15]. In the specific case of Wasserstein barycenters, dimensionality reduction has several practical and theoretical benefits, including lower storage space, faster running time in computing distances, and versatility: it can be used as a pre-processing tool and combined with *any* algorithm for computing the Wasserstein barycenter.

## 1.1 Our Results

In this paper, we study dimensionality reduction techniques for computing a Wasserstein barycenter of discrete probability distributions. Our main results show that it is possible to project the distributions into low dimensions while provably preserving the quality of the barycenter. A key result in dimensionality reduction is the classical Johnson-Lindenstrauss (JL) lemma [JL84], which states that projecting a dataset of $N$ points into roughly $O(\log N)$ dimensions is enough to preserve all pairwise distances.

Using the JL lemma, we first show that we can assume the distributions lie in $O(\log(nk))$ dimensions, where $k$ is the number of input distributions whose barycenter we are computing, $n$ is the size of the support of the barycenter, and each of the $k$ input distributions has support size $\text{poly}(n)$. For $p = 2$, there exists a closed form for the cost of any candidate barycenter in terms of the pairwise distances of the points in the input distributions. Thus it is straightforward to see that our bound results from the fact that there are $k \cdot \text{poly}(n)$ total points masses in the union of all the distributions and therefore, projecting them into a dimension of size $O(\log(k \, \text{poly}(n))) = O(\log(nk))$ suffices to preserve all of their pairwise distances. However for $p \neq 2$, a closed form for the optimal cost no longer exists, so preservation of all pairwise distances is insufficient. Instead, we make use of a Lipschitz extension theorem, namely the Kirszbraun theorem, which allows us to "invert" the dimensionality reduction map and argue the preservation of the cost of the Wasserstein barycenter under a general $L_p$ objective. For more details, see Section 3.

**Dimensionality reduction independent of** $k$**.** While the JL lemma is known to be tight [LN16, LN17], it is possible to improve its dimensionality guarantees for *specific* problems, such as various formulations of clustering [CEM$^+$15, BBC$^+$19, MMR19]. Indeed, our main result is that we can achieve a dimension bound *beyond* the $O(\log(nk))$ bound that follows from the JL lemma and Kirszbraun theorem. We show that it suffices to project the support points onto $O(\log n)$ dimensions, which is *independent* of the number of distributions $k$. In fact, we show a stronger statement that projecting the points supported by the distributions onto $O(\log n)$ dimension preserves the cost of the objective (1) for *any* distribution $\nu$ supported on at most $n$ points (Theorem 4.1). The algorithmic application of this theorem is that one can take *any* approximation algorithm or heuristic for computing the Wasserstein barycenter and combine it with dimensionality reduction. A simplification of our theorem is stated below where we omit some parameters for clarity.

**Theorem 1.1** (Theorem 4.1 Simplified)**.** *Let $\mu_1, \ldots, \mu_k$ be discrete probability distributions on $\mathbb{R}^d$ such that $|supp(\mu_i)| \leq \text{poly}(n)$ for all $i$. There exists a dimensionality reduction map $\pi : \mathbb{R}^d \to \mathbb{R}^m$ for $m = O(\log n)$ such that projection under $\pi$ preserves the cost of objective (1) for any $\nu$ supported on at most $n$ points.*

The result is surprising because the projected dimension is independent of the number of input distributions $k$, which could be significantly larger than $n$. Thus the random projection map $\pi$ can no longer even guarantee the preservation of a significant fraction of pairwise distances between the support points of the $k$ distributions. Our main tool is a "robust" Lipschitz extension theorem introduced in [MMR19] for $k$-means clustering. We adapt this analysis to the geometry of the Wasserstein barycenter problem.

**Optimality of dimensionality reduction.** We complement our upper bound results by showing that our dimension bound of $\log n$ dimensions is *tight* if a random Gaussian matrix is used as the projection map. We also show that the JL lemma is tight for the related problem of computing the optimal transport between two distributions with support of size $n$. More specifically, we give a lower bound showing that $\Omega(\log n)$ dimension is needed for a random projection to preserve the optimal

transport cost. Thus our results show a separation between the geometry of the optimal transport problem and the geometry of the Wasserstein barycenter problem, as we overcome the JL bound in the latter.

**Hardness of approximation.** In addition, we also show the NP-hardness of approximation for the Wasserstein barycenter problem. Namely, we show that it is NP-hard to find an approximate barycenter that induces a cost that is within a factor of $1.0013$ of the optimal barycenter if we restrict the support size of the barycenter. This complements recent work of [AB21, BP21], who showed that computing sparse Wasserstein barycenters is NP-hard.

**Coresets for Wasserstein barycenters.** An alternate way to reduce the complexity of datasets is through the use of coresets, which decrease the effective data size by reducing the number of input points rather than the input dimension $d$. If the number of input distributions $k$ is significantly larger than the support size $n$, we show that there exists a weighted subset $C$ of roughly $\text{poly}(n)$ distributions, so that computing the optimal barycenter on $C$ is equivalent to computing the optimal barycenter on the original input up to a small approximation loss. Hence, it can potentially be much more efficient to use the subset $C$ in downstream algorithms involving Wasserstein barycenters. Moreover, the coreset is not mutually exclusive with our techniques for reducing the ambient dimension $d$. Our techniques show that we can simultaneously reduce both the size of the input distribution $k$ and the dimension $d$ of the data, while preserving the optimal clustering within a small approximation factor.

In Supplementary Section E, we also show a connection between the Wasserstein barycenter problem and constrained low-rank problems. This class of problems includes examples such as the singular value decomposition (SVD) and $k$-means clustering. While this connection does not yield any improved results, it classifies the Wasserstein barycenter as a member of a general class of problems, and this classification could have further applications in the future.

**Experiments.** Finally, we present experimental evaluation of our proposed methodology. Note that our results imply that we can use dimensionality reduction in conjunction with *any* Wasserstein barycenter algorithm and still roughly retain the approximation guarantees of the algorithm used. Specifically, we give examples of real high dimensional datasets such that solving the Wasserstein barycenter problem in a reduced dimension leads to computational savings while preserving the quality of the solution. Our experiments in Section 7 demonstrate that on natural datasets, we can reduce the dimension by 1-2 orders of magnitude while increasing the solution cost by only $5\%$. We also empirically test our coreset construction. Our method both reduces error and requires fewer samples than simple uniform sampling.

## 1.2 Related Work

[AB21, BP21] showed that computing sparse Wasserstein barycenters is NP-hard; hence, most of the algorithmic techniques focus on computing approximate Wasserstein barycenters that induce a cost within an additive $\varepsilon$ of the optimal cost. [AC11b] first considered approximating Wasserstein barycenters when either (1) the distributions $\mathbb{P}$ only have discrete support on $\mathbb{R}$, (2) $k = 2$, or (3) the distributions $\mu_i$ are all multivariate Gaussians in $\mathbb{R}^d$. Although there is a line of research that studies the computation of barycenters of continuous distributions, e.g. [ÁDCM16, CMRS20], we focus on discrete input distributions. For discrete input distributions, the majority of the literature can be categorized by its assumptions of the support of the barycenter [ABA21].

**Fixed-support.** The "fixed-support approximation" class of algorithms assume that the support of the barycenter is among a fixed set of possible points. It then remains for the algorithms to solve a polynomial-size linear program associated with the corresponding set [CD14, BCC+15, COO15, SCSJ17, KTD+19, LHC+20]. Unfortunately, the set of possible points must often be an $\varepsilon$-net over the entire space, which results in a size proportional to $1/\varepsilon^d$ that suffers from the curse of dimensionality. Nevertheless for constant dimension, the algorithms typically have runtime $\text{poly}(n, k, D/\varepsilon)$, where $D$ is an upper bound on the diameter of the supports of the input distributions. This is further improved by an algorithm of [ABA21] that achieves runtime $\text{poly}(n, k, \log(D/\varepsilon))$.

**Free support.** A separate class of algorithms do not make assumptions about the possible support of the optimal barycenter. These "free-support algorithms" instead optimize over the entire set of up candidate barycenters, which can be as large as $n^k$ in quantity. Thus these algorithms, e.g., [CD14, LSPC19], either use exponential runtime or a number heuristics that lack theoretical guarantees. [ABA21] showed how to explore the $n^k$ possible points in polynomial time for fixed $d$.

## 2 Preliminaries

**Notation.** For a positive integer $n$, we denote $[n] := \{1, 2, \ldots, n\}$. We use $\mu_1, \ldots, \mu_k$ to denote the $k$ distributions whose Wasserstein barycenter we wish to compute. While the Wasserstein barycenter problem is well defined for continuous distributions, in practice and in actual computations, the distributions $\mu_i$ are assumed to be discrete distributions that are supported on some number of point masses. This is also the assumption we make. More specifically, we assume that each of the distributions $\mu_i$ are discrete distributions supported on at most $T \leq n^C$ points where $C$ is a fixed constant. That is, $\mu_i = \sum_{j=1}^{T} a(x_{ij})\delta_{x_{ij}}$, where $\delta_x$ is a delta function at $x$ and $a(x)$ is the weight assigned to a point $x$ in its corresponding $\mu_i$. We note that if there is some point $x$ in the support of more than one of the $\mu_i$s, then the weight function $a$ may not be well-defined. Instead, we implicitly assume that $a = a(x, i)$ is a function of both the point *and* the distribution from which it comes, but we suppress this dependence on $i$ for notational clarity.

The distribution $\nu$ denotes a candidate for the Wasserstein barycenter of the $\mu_i$. We write $\nu = \sum_{j=1}^{n} b_j \delta_{\nu^j}$. In general, an actual Wasserstein barycenter (in the sense of minimizing the objective (1) over all possible $\nu$ of any support size) may have support size up to $|\bigcup_{i=1}^{k} \text{supp}(\mu_i)|$ [ABM16]. Throughout this paper, we will restrict ourselves to computing (approximate) barycenters of support size at most $n$. When we refer to an optimal barycenter, we mean a distribution that minimizes the objective (1) *within this restricted class*.

**Problem description.** The goal is to compute a distribution $\nu \in \mathbb{R}^d$, consisting of at most $n$ point masses, to minimize the objective (1). As previously mentioned, $W_p(\mu_i, \nu)$ is the Wasserstein $p$-metric, defined as

$$W_p(\mu, \nu) = \inf_{\gamma \in \Gamma(\mu,\nu)} \left( \int_{\mathbb{R}^d \times \mathbb{R}^d} \|x - y\|^p d\gamma(x, y) \right)^{1/p}$$

where $\Gamma(\mu, \nu)$ is the set of all joint distributions with marginals $\mu$ and $\nu$ (i.e. all couplings of $\mu$ and $\nu$) and $\|\cdot\|$ denotes the Euclidean norm on $\mathbb{R}^d$. When $\mu$ and $\nu$ are discrete distributions, $W_p(\mu, \nu)^p$ $p$-metric is the cost of the minimum cost flow from $\mu$ to $\nu$ with edge costs being the Euclidean distance raised to the $p$-th power. For simplicity, we assume that the distributions $\mu_1, \cdots, \mu_k$ are weighted equally (each $\lambda_i = 1/k$ in (1)) but our results hold in the general case as well. The most common choice of $p$ is $p = 2$.

**Description of $\nu$.** The barycenter $\nu$ can be characterized as follows. Recall that $\nu$ is supported on the points $\nu^1, \ldots, \nu^n$. For the optimal coupling of each $\mu_i$ to $\nu$, let $w_j(x)$ denote the total weight sent from $x$ (in the support of one of the $\mu_i$s) to $\nu^j$. (The same note about suppressing the dependence of $w_j$ on the distribution $\mu_i$ from which $x$ comes applies here.) Let $S_j = \{x \in \bigcup_{i=1}^{k} \text{supp}(\mu_i) : w_j(x) > 0\}$ denote the set of all points in the $\mu_i$s with some weight sent to $\nu^j$. Then *given* the set $S_j$ and weighting function $w_j(\cdot)$, we can reconstruct $\nu^j$ since it must minimize the objective

$$\sum_{x \in S_j} w_j(x)\|x - \nu^j\|^p. \tag{2}$$

Indeed if $\nu^j$ does not minimize this quantity, we can change it and reduce the cost of (1).

Consider the case of $p = 2$. For a fixed $j$, (2) is just a weighted $k$-means problem whose solution is the weighted average of the points in $S_j$. To prove this, consider taking the gradient of (2) with respect to the $k$-th coordinate of $\nu^j$. Then setting it equal to 0 gives us that the $k$-th coordinate will be the weighted average of the $k$-th coordinates of the points $S_j$. That is, we have

$$\nu^j = \frac{\sum_{x \in S_j} w_j(x)x}{\sum_{x \in S_j} w_j(x)} = \frac{1}{kb_j} \sum_{x \in S_j} w_j(x)x. \tag{3}$$

The second equality results from observing that in order for the $w_j$s to define a proper coupling, we have $\sum_{j=1}^{n} w_j(x) = a(x)$ for all $x$ in the support of the $\mu_i$s, and $\sum_{x \in \text{supp}(\mu_i)} w_j(x) = b_j$ for all $i$, along with $w_j(x) \geq 0$. In particular, this implies that $\sum_{x \in S_j} w_j(x) = kb_j$ for all $j = 1, \ldots, n$.

For arbitrary $p$, such a concise description of $\nu^j$ is not possible. Therefore an alternate, but equivalent, way to characterize the distribution $\nu$ is to just define the sets $S_j$ and weight functions $w_j(\cdot)$ for $1 \leq j \leq n$. This motivates the following definitions.

**Definition 2.1.** *A solution $(S, w) = (S_1, \ldots, S_n, w_1, \ldots, w_j)$ is a valid partition as described previously (meaning that these partitions come from the optimal coupling between each $\mu_i$ to a fixed $\nu$), along with the corresponding weight functions $w_j(\cdot)$.*

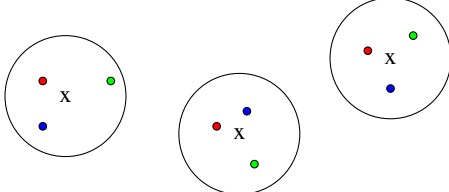

Figure 1: Points of the same color belong to the same distribution. The sets $S_j$ are denoted by the large black circles. Given the partitions $S_j$ (denoted by large black circles) and associated weight functions $w_j$, we can reconstruct the barycenter (denoted by crosses).

**Definition 2.2.** *Let $(S, w)$ be a solution. The cost of this solution, denoted $\text{cost}_p(S)$, is the value of the objective (1) when we reconstruct $\nu$ from $S$ and $w$ and evaluate (1):*

$$\text{cost}_p(S) = \min_{\nu} \frac{1}{k} \sum_{j=1}^{n} \sum_{x \in S_j} w_j(x) \|x - \nu^j\|^p.$$

*Similarly for a projection $\pi$, $\text{cost}_p(\pi S)$ denotes the value of the objective (1) when we first project each of the distributions to $\mathbb{R}^m$ using $\pi$, then compute $\widetilde{\nu}$ using the original weights $w_j$:*

$$\text{cost}_p(\pi S) = \min_{\widetilde{\nu}} \frac{1}{k} \sum_{j=1}^{n} \sum_{x \in S_j} w_j(x) \|\pi(x) - \widetilde{\nu}^j\|^p.$$

*Note that each $\widetilde{\nu}^j \in \mathbb{R}^m$. We suppress the dependence of the cost on $w$ for notational convenience.*

For the case of $p = 2$, we can further massage the value of $\nu^j$ in (2). Let $\bar{x}$ denote the weighted average of points in $S_j$ (given by (3)). From our discussion above, we know that $\nu^j = \bar{x}$. After some standard algebraic manipulation, we can show that $\sum_{x \in S_j} w_j(x) \|x - \nu^j\|^2 = \sum_{x \in S_j} w_j(x) \|x\|^2 - k b_j \|\bar{x}\|^2$ and $\sum_{x,y \in S_j} w_j(x) w_j(y) \|x - y\|^2 = 2 k b_j \left( \sum_{x \in S_j} w_j(x) \|x\|^2 - k b_j \|\bar{x}\|^2 \right)$. Combining these equations yields the following for the $p = 2$ objective.

$$\frac{1}{2 k b_j} \left( \sum_{x,y \in S_j} w_j(x) w_j(y) \|x - y\|^2 \right) = \sum_{x \in S_j} w_j(x) \|x - \nu^j\|^2. \tag{4}$$

**Dimension reduction.** In this paper we are concerned with dimensionality reduction maps $\pi : \mathbb{R}^d \to \mathbb{R}^m$ that are JL projections, i.e., any dimensionality reduction map that satisfies the condition of the JL lemma. This includes random Gaussian and sub-Gaussian matrices [LN16, MMR19]. We are mainly concerned with making the projection dimension $m$ as small as possible.

Consider any algorithm $\mathcal{A}$ that, given $\mu_1, \cdots, \mu_k$, solves for some approximate or exact $\nu$ minimizing the objective (1). We can combine any such $\mathcal{A}$ with dimensionality reduction by first projecting the point masses of the $\mu_i$ down to $\mathbb{R}^m$ for some $m < d$ and using $\mathcal{A}$ to compute some barycenter $\widetilde{\nu}$ in $\mathbb{R}^m$. Then, we can consider the solution $(S, w)$ induced by $\widetilde{\nu}$ (see Definitions 2.1 and 2.2) to *reconstruct* the appropriate $\nu$ in the original dimension $\mathbb{R}^d$ using the objective Eq. (2). Note that this objective is a convex program for any $p \geq 1$ since we are given $S_j$ and $w_j(\cdot)$. For $p = 2$ (which is the most common case), $\nu^j$ has a particularly simple form which is the weighted average of the points in $S_j$ (see Eq. (3)). This procedure is outlined in Algorithm 1.

As a corollary of our results, if algorithm $\mathcal{A}$ takes time $T(n, k, d)$, then using dimensionality reduction as in the procedure outlined above takes time $T(n, k, m)$ plus the time to perform the projection and reconstruct the barycenter using the solution $S$. The cost of running algorithm $\mathcal{A}$ is usually much more expensive than performing the projection, and the reconstruction step can also be solved efficiently since it is convex. In the case of $p = 2$, the reconstruction just amounts to computing $n$ weighted means. Therefore for $m \ll d$, we get significant savings since $T(n, k, m) \ll T(n, k, d)$.

---
**Algorithm 1** Using dimensionality reduction with any algorithm $\mathcal{A}$ for computing WB
---
**Require:** $k$ discrete distributions $\mu_1, \cdots, \mu_k$ with point masses in dimension $\mathbb{R}^d$, projection dimension $m$, algorithm $\mathcal{A}$
 1: Project the point masses of each distribution $\mu_i$ to dimension $\mathbb{R}^m$ using a JL projection
 2: Use algorithm $\mathcal{A}$ to solve (or approximately solve) the Wasserstein barycenter problem in $\mathbb{R}^m$ to get a distribution $\widetilde{\nu}$    //$\widetilde{\nu}$ is a discrete distribution in $\mathbb{R}^m$
 3: Let $(S, w)$ be the solution that partitions the the point masses of the distributions as described in Definition 2.1
 4: **for** each $S_j \in S$ **do**
 5:    Solve for $\nu^j$ minimizing $\sum_{x \in S_j} w_j(x) \|x - \nu^j\|^p$    //This is a convex program for $p \geq 1$. For $p = 2$, $\nu^j$ is just the weighted average of points in $S_j$.
 6: **end for**
 7: Output the distribution $\nu$ supported on $\nu^j$, and where $\nu^j$ has the same weight as $\widetilde{\nu}^j$
---

# 3    Reduction to $O(\log(nk))$ Dimensions

We first show that it suffices to project the point masses of the input distribution into $O(\log(nk))$ dimensions and guarantee that the cost of any *solution* is preserved. Note that our results hold *simultaneously for all solutions.* We first state the $p = 2$ case.

**Theorem 3.1.** *Consider a JL projection $\pi$ from $\mathbb{R}^d$ to $\mathbb{R}^m$ for $m = O(\log(nk/\delta)/\varepsilon^2)$. Then*

$$\mathbb{P}\left(\text{cost}_2(\pi S) \in \left[(1 - \varepsilon)^2 \cdot \text{cost}_2(S), (1 + \varepsilon)^2 \cdot \text{cost}_2(S)\right] \text{for all solutions } S\right) \geq 1 - \delta.$$

*Proof.* The proof follows from the solution decomposition given in (4) if we condition on all the pairwise distances being preserved which happens with probability $1 - \delta$.    $\square$

A decomposition similar to (4) does not exist for $p \neq 2$. To prove an analogous theorem for $p \neq 2$, we need the following Lipschitz extension theorem which roughly allows us to "invert" a dimensionality reduction map.

**Theorem 3.2** (Kirszbraun Theorem [Kir34])**.** *For any $D \subset \mathbb{R}^m$, let $f : D \to \mathbb{R}^d$ be an L-Lipschitz function. Then there exists some extension $\widetilde{f} : \mathbb{R}^m \to \mathbb{R}^d$ of $f$ to the entirety of $\mathbb{R}^m$ such that $f(x) = \widetilde{f}(x)$ for all $x \in D$ and $\widetilde{f}$ is also L-Lipschitz.*

The Kirszbraun theorem allows us to prove Theorem 3.1 for general $p$ with a dimension bound of $m = O(\log(nk/\delta)p^2/\varepsilon^2)$ (see Theorem A.1 in Supplementary Section A).

The overview for the proof strategy for the general $p \neq 2$ case is as follows. First suppose that all the pairwise distances between the support points of all the distributions are preserved under the projection map up to multiplicative error $1 \pm \varepsilon$. This event happens with probability at least $1 - \delta$. We then consider the map $f : \mathbb{R}^m \to \mathbb{R}^d$ that maps each of the *projected* points to its original counterpart in $\mathbb{R}^d$. Note that the map is from the *smaller* dimension $m$ to the larger dimension $d$. On the support points, we know that $f$ is $(1 + \varepsilon)$-Lipschitz by our assumption above.

Now if the projection caused the cost of $\pi S$ to decrease significantly, then using the Kriszbraun theorem, one could "lift" the corresponding barycenter $\widetilde{\nu}$ from the projected dimension to the original dimension using the extension map $\widetilde{f}$. Then since $\widetilde{f}$ is Lipschitz, this lifted barycenter $\widetilde{f}(\widetilde{\nu})$ plugged into Eq. (2) would subsequently have cost smaller than the original barycenter that corresponds $S$ in the original dimension. This is a contradiction in light of Eq. (2) and the description of $\nu$ given in Section 2. Note that the exact description of $\widetilde{f}$ does not matter for the analysis, just that such a map exists. A complete, rigorous proof can be found in the supplementary section.

# 4    Optimal Dimensionality Reduction

We now present our main theorem which improves the guarantees of Theorems 3.1 and A.1.

**Theorem 4.1.** *Let $\mu_1, \ldots, \mu_k$ be discrete probability distributions on $\mathbb{R}^d$ such that $|supp(\mu_i)| \leq$ poly$(n)$ for all $i$. Let $d \geq 1$, $\varepsilon, \delta \in (0, 1)$, and $p \geq 1$. Let $\pi_{d,m} : \mathbb{R}^d \to \mathbb{R}^m$ be a family of random JL maps with $m = O\left(\frac{p^4}{\varepsilon^2} \log \frac{n}{\varepsilon \delta}\right)$. Then we have,*

$$\mathbb{P}\left(\mathrm{cost}_p(\pi S) \in \left[(1 - \varepsilon) \cdot \mathrm{cost}_p(S), (1 + \varepsilon) \cdot \mathrm{cost}_p(S)\right] \text{ for all solutions } S\right) \geq 1 - \delta.$$

We now give an overview of the proof strategy for Theorem 4.1, deferring all technical details to the supplementary section. Ideally, one would like to use a strategy similar to the proof of Theorem A.1. The key bottleneck is that when we project down to the $m$ specified in Theorem 4.1, a large number of pairwise distances between the support points of the $k$ distributions can be distorted (since we are projecting to a dimension smaller than $O(\log(nk))$). Therefore, the Kirszbraun theorem cannot apply as the map $f$ described in the proof strategy of Theorem A.1 is no longer Lipschitz on the support points.

To overcome this barrier, we generalize an approach of [MMR19], who achieved the optimal dimensionality bounds for $k$-means clustering beyond the naïve JL bound by defining a *distortion graph* on the set of input points, which has an edge between each pair of points if their pairwise distance is distorted by at least a $(1 + \varepsilon)$-factor under the random projection map $\pi$. They show that the distortion graph is everywhere sparse, i.e., each vertex has small expected degree in the distortion graph, which implies a "robust" Kirszbraun theorem (for their particular problem of $k$-means clustering). Namely, there exists an extension map $\widetilde{f} : \mathbb{R}^d \to \mathbb{R}^m$ and a specific point $v \in \mathbb{R}^m$ in the projected space such that a large fraction of the distances from the pre-image $\widetilde{f}^{-1}(v)$ to the input points in $\mathbb{R}^d$ are preserved. Moreover, the input points whose distance to $\widetilde{f}^{-1}(v)$ is not preserved can be shown to contribute small error to the $k$-means clustering cost.

The dimensionality reduction maps of Theorem 4.1 generally require multiplication by a dense matrix of (scaled) subgaussian variables. In the Supplementary Section, we show that "faster" dimensionality reduction maps can also be used by providing a trade off between the projection runtime and the dimension $m$. Note that in practice, performing the projection is extremely cheap since we only need to perform one matrix multiplication, which is highly optimized. Therefore the cost of any algorithm for Wasserstein barycenter will typically outweigh the cost of computing the projection.

## 4.1 Dimensionality Reduction Lower Bounds

In this section, we state lower bounds on the projection dimension $m$ for the Wasserstein barycenter problem. Theorem 4.2 shows that Theorem 4.1 is tight up to constant factors.

**Theorem 4.2.** *Consider the setup of Theorem 4.1. Any Gaussian matrix used as a dimension reduction map that allows a $(1 + \varepsilon)$-approximation to the optimal Wasserstein barycenter requires dimension $\Omega(\log n / \varepsilon^2)$.*

We also prove that one cannot do better than the naïve JL bound for the related problem of computing the optimal transport between two discrete distributions with $n$ point masses each. This is in contrast to the case of Wasserstein barycenter where we were able to overcome the bound that comes from the JL lemma alone. Theorem 4.3 shows that the optimal solution in the projected dimension can induce a poor quality solution in the original dimension if the projection dimension is smaller than $\log n$.

**Theorem 4.3.** *There exists point sets $A, B \subset \mathbb{R}^d$ with $|A| = |B| = n$ and matching cost $M$ between them, such that if randomly projected down to $m = o(\log n)$ dimensions using an appropriately scaled Gaussian random matrix, the pull back cost of the optimal matching in $\mathbb{R}^m$ is at least $\omega(M)$.*

In addition, we prove a related theorem which states that the *cost* of the optimal transport is heavily distorted if we project to fewer than $\log n$ dimensions.

**Theorem 4.4.** *There exists point sets $A, B \subset \mathbb{R}^d$ with $|A| = |B| = n$ and matching cost $M$ between them, such that if randomly projected down to $m = o(\log n)$ dimensions using an appropriately scaled Gaussian random matrix, the cost of optimal matching in $\mathbb{R}^m$ is $o(M)$ with probability at least $2/3$.*

See Supplementary Section D for full proofs.

# 5 Coresets

In this section, we give a coreset construction for Wasserstein barycenters. Our goal is to reduce the number of distributions $k$ to only depend polynomially on $n$. We first define our notion of coresets.

**Definition 5.1** (Coreset). *Fix $p \geq 1$. Let $C$ and $M$ be two sets of distributions in $\mathbb{R}^d$ where all distributions consist of* $\mathrm{poly}(n)$ *point masses. $C$ is called an $\varepsilon$-corset for the set of distributions $M$ if there exist weights $w_c$ for $c \in C$ such that for all distributions $\nu$ of support size at most $n$, it holds that*

$$(1 - \varepsilon) \sum_{c \in C} w_c \, W(c, \nu)^p \leq \frac{1}{|M|} \sum_{\mu \in M} W(\mu, \nu)^p \leq (1 + \varepsilon) \sum_{c \in C} w_c \, W(c, \nu)^p.$$

The main result of this section is the following theorem.

**Theorem 5.2** (Theorem C.6 simplified). *Let $M$ be a set of discrete distributions in $\mathbb{R}^d$, each supported on at most $\mathrm{poly}(n)$ point masses. There exists a weighted subset $K \subseteq M$ of size $\mathrm{poly}(n, d)/\varepsilon^2$ that satisfies Definition 5.1 for $p = O(1)$.*

To prove Theorem 5.2, we follow the "importance sampling" by sensitivities framework in conjunction with using structural properties of the Wasserstein barycenter problem itself. The sensitivity sampling framework has been successfully applied to achieve corsets for many problems in machine learning (see the references in the survey [BLK17]). Note that we have not attempted to optimize the constants in our proofs and instead focus on showing that $k$ can be reduced to $\mathrm{poly}(n, d)$ for simplicity. The formal proof of Theorem 5.2 is deferred to the supplementary section.

We now describe the high level overview of the proof. We form the set $C$ by sampling distributions in $M$ with replacement based on their "importance" or contribution to the total cost. The notion of importance is formally captured by the definition of sensitivity.

**Definition 5.3** (Sensitivity). *Consider the set $N$ of all possible barycenter distributions $\nu$ with support size at most $n$. The sensitivity of a distribution $\mu \in M$ is defined as*

$$\sigma(\mu) = \sup_{\nu \in N} \frac{W(\mu, \nu)^p}{\frac{1}{|M|} \sum_{\mu' \in M} W(\mu', \nu)^p}.$$

*The total sensitivity is defined as $\mathfrak{S} = \frac{1}{|M|} \sum_{\mu \in M} \sigma(\mu)$.*

To see why such a notion is beneficial, consider the case that one distribution $\mu$ consists of point masses that are outliers among all of the point masses comprising the distributions in $M$. Then it is clear that we must sample $\mu$ with a higher probability if we wish to satisfy the definition of a coreset. In particular, we sample each distribution in $M$ with probability proportional to (an upper bound on) its sensitivity. Using a standard result in coreset construction, we can bound the size of the coreset in terms of the total sensitivity and a measure of the "complexity" of the Wasserstein barycenter problem which is related to the VC dimension. In particular, we utilize the notion of psuedo-dimension.

**Definition 5.4** (Pseudo-Dimension, Definition 9 [LFKF18]). *Let $\mathcal{X}$ be a ground set and $\mathcal{F}$ be a set of functions from $\mathcal{X}$ to the interval $[0, 1]$. Fix a set $S = \{x_1, \cdots, x_n\} \subset \mathcal{X}$, a set of reals numbers $R = \{r_1, \cdots, r_n\}$ with $r_i \in [0, 1]$ and a function $f \in \mathcal{F}$. The set $S_f = \{x_i \in S \mid f(x_i) \geq r_i\}$ is called the induced subset of $S$ formed by $f$ and $R$. The set $S$ with associated values $R$ is shattered by $\mathcal{F}$ if $|\{S_f \mid f \in \mathcal{F}\}| = 2^n$. The pseudo-dimension of $\mathcal{F}$ is the cardinality of the largest shattered subset of $\mathcal{X}$ (or $\infty$).*

The following theorem provides a formal connection between the size of coresets and the notion of sensitivity and psuedo-dimension. Note that the statement of the theorem is more general and applies to a wider class of problems. However, we specialize the theorem statement to the case of Wasserstein Barycenters.

**Theorem 5.5** (Coreset Size, Theorem 2.4.6 in [Lan18], Theorem 2.3 in [BLK17] for the case of Wasserstein Barycenters). *Let $\varepsilon > 0$ and $\delta \in (0, 1)$. Let $s : M \to \mathbb{R}^{\geq 0}$ denote any upper bound function on the sensitivity $\sigma(\cdot)$ defined in Definition 5.3 and let $S = \frac{1}{|M|} \sum_{\mu \in M} s(\mu)$. Consider a set $K$ of $|K|$ samples of $M$ with replacement where each distribution $\mu \in M$ is sampled with probability $q(\mu) = s(\mu)/(|M| \cdot S)$ and each sampled point is assigned the weight $1/(|M| \cdot |K| \cdot q(\mu))$. Let $\mathcal{F}$*

*denote the set of functions*

$$\mathcal{F} = \left\{ \frac{W(\cdot, \nu)^p}{Sq(\cdot) \sum_{\mu \in M} W(\mu, \nu)^p} \mid \nu \in N \right\}$$

*where $N$ is the set of all possible barycenter distributions with support size at most $n$. Let $d'$ denote the pseudo-dimension of $\mathcal{F}$. Then the set $K$ (along with the associated weights) satisfies Definition 5.1 with probability at least $1 - \delta$ if*

$$|K| \geq \frac{cS}{\varepsilon^2} \left( d' \log S + \log \frac{1}{\delta} \right)$$

*where $c > 0$ is some absolute constant.*

Thus, the bulk of our work lies in bounding the sensitivities and psuedo-dimension. For the former quantity, we exploit the fact that the Wasserstein distance is a metric. The latter requires us to use tools from statistical learning theory which relate the VC dimension of a function class to its algorithmic complexity (see Lemmas C.3 and C.4). Full details given in Supplementary section C.

## 6 Other Theoretical Results

We now present some additional theoretical results pertaining to Wasserstein barycenters. Our first result is that Wasserstein barycenters can be formulated as a constrained low-rank approximation problem. This class of problems includes coputing the SVD and $k$-means clustering [CEM$^+$15]. Formally, we prove the following theorem.

**Theorem 6.1.** *Given discrete distributions $\mu_1, \ldots, \mu_k \in \mathbb{R}^d$ with support size at most $n$, consider the problem of computing the Wasserstein barycenter with support size at most $n$ for the $p = 2$ objective. There exists a matrix $A \in \mathbb{R}^{nk \times d}$ and a set $S$ of rank $n$ orthogonal projection matrices in $\mathbb{R}^{nk \times nk}$ such that the first problem is equivalent to computing*

$$\boldsymbol{P}^* = \underset{\boldsymbol{P} \in S}{\operatorname{argmin}} \|\boldsymbol{A} - \boldsymbol{P}\boldsymbol{A}\|_F^2.$$

The proof of Theorem 6.1 is given in Section E.

We also prove the following NP hardness result in Section F which complements the hardness results in [AB21, BP21].

**Theorem 6.2.** *It is NP-hard to approximate an optimal Wasserstein barycenter of fixed support size up to a multiplicative factor* 1.0013.

## 7 Experiments

In this section, we empirically verify that dimensionality reduction can provide large computational savings without significantly reducing accuracy. We use the following datasets in our experiments.

**FACES dataset**: This dataset is used in the influential ISOMAP paper and consists of 698 images of faces in dimension 4096 [TSL00]. We form $k = 2$ distributions by splitting the images facing to the "left" versus the ones facing "right." This results in $\sim 350$ uniform point masses per distribution.

**MNIST dataset**: We subsample $10^4$ images from the MNIST test dataset (dimension 784). We split the images by their digit class which results in $k = 10$ distributions with $\sim 10^3$ uniform point masses each in $\mathbb{R}^{784}$.

**Experimental setup.** We project our datasets in dimensions $d$ ranging from $d = 2$ to $d = 30$ and compute the Wasserstein barycenter for $p = 2$. For FACES, we limit the support size of the barycenter to be at most 5 points in $\mathbb{R}^{4096}$ (since the barycenter should intuitively return an "interpolation" between the left and right facing faces, it should not be supported on too many points). For MNIST we limit the support size of the barycenter to be at most 40. We then take the barycenter found in the lower dimension and compare its cost in the higher dimension (see Algorithm 1) against the Wasserstein barycenter found in the higher dimension.

We use the code and default settings from [Ye19] to compute the Wasserstein barycenter; this implementation has been applied in previous empirical papers [YWWL17]. While we fix this implementation, note that dimensionality reduction is extremely flexible and can work with any algorithm or implementation (see Algorithm 1) and we would expect it to produce similar results.

**Results.** Our results are displayed in Figure 2. We see that for both datasets, reducing the dimension to $d = 30$ only increases the cost of the solution by $5\%$. This is **1-2** orders of magnitude smaller than from the original dimensions of 784 and 4096 for MNIST and FACES respectively. The average time taken to run the Wasserstein barycenter computation algorithm in $d = 30$ was $73\%$ and $9\%$ of the time taken to run in the full dimensions respectively.

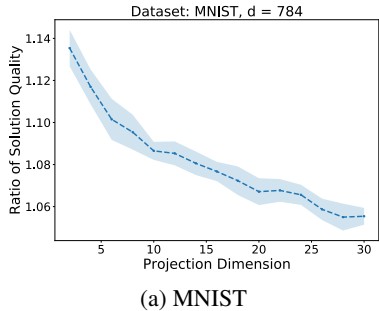
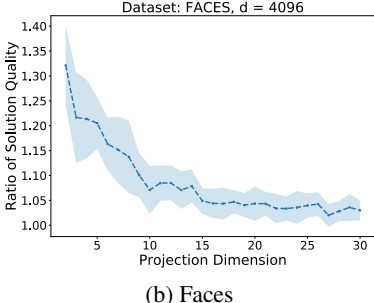

(a) MNIST                       (b) Faces

Figure 2: Ratio of the quality of solution found in the lower dimension versus the original dimension. Result displays average of 20 independent trials and $\pm 1$ standard deviation is shaded.

**Coreset experiments.** Our coreset result reduces the number of distributions $k$ through sensitivity (importance) sampling. We created a synthetic dataset with large $k$ but small $n$ and $d$ to emphasize the advantage of sensitivity sampling over uniform sampling. We have $k = 50,000$ distributions that each consists of a single point mass in $\mathbb{R}$. The first $k - 1$ distributions are all supported at the origin while one distribution is supported at $x = k$. We consider the $p = 2$ case and limit the support size of the barycenter to also be 1. Let $\text{cost}_{\text{orig}}(\nu)$ denote the cost of $\nu$ on the original objective (1) and let $\text{cost}_{\text{core}}(\nu)$ the cost of (1) when evaluated on a coreset. We record the relative error $|\text{cost}_{\text{core}}(\nu) - \text{cost}_{\text{orig}}(\nu)|/|\text{cost}_{\text{orig}}(\nu)|$ evaluated at $\nu = \delta_x$, i.e. a single unit point mass at $x$, for $x = 0, 1, 10$. We then average the results across 10 trials each. As $x$ (the point on which the query distribution is supported) grows bigger, the associated cost became bigger, hence decreasing the relative error. Other query locations displayed the same trend. See Figure 3 for more details.

| Method | # of samples | % error at query | | | |
|---|---|---|---|---|---|
| | | $x = 100$ | $x = 10$ | $x = 1$ | $x = 0$ |
| Uniform sampling | 1000 | 0.986 | 9.087 | 49.998 | 100 |
| Sensitivity sampling | 10 | 0.0040 | 0.0036 | 0.0020 | 0 |

Figure 3: Even with much fewer samples, sensitivity sampling outperforms uniform sampling for a number of query locations, averaged across 10 repetitions.

# Acknowledgments

Sandeep Silwal was supported in part by a NSF Graduate Research Fellowship Program. Samson Zhou was supported by a Simons Investigator Award of David P. Woodruff.

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
