$$\mathbb{P}\left(\text{cost}_p(\pi S) \in [(1-\varepsilon)\cdot\text{cost}_p(S),(1+\varepsilon)\cdot\text{cost}_p(S)] \text{ for all solutions } S\right) \geq 1 - \delta,$$

*where the probability is taken over the randomness in the projection $\pi$.*

*Proof of Theorem A.1.* We again assume that the distances between point masses among all the $nk$ points in the distributions $\mu_1,\ldots,\mu_k$ are preserved up to $1 \pm \varepsilon$. By Theorem 3.2, dimensionality reduction gives us a $(1+\varepsilon)$-Lipschitz map $\widetilde{\pi} : \mathbb{R}^d \to \mathbb{R}^m$ as well as $\widetilde{\varphi} : \mathbb{R}^m \to \mathbb{R}^d$.

Now consider an arbitrary solution $S = (S_1,\ldots,S_n)$. We first show that $\text{cost}(S) \leq (1+\varepsilon)^p\text{cost}(\pi S)$ where $\text{cost}(\pi S)$ is the cost of the solution $S$ evaluated in the projected space. Indeed for any $S_j$, the objective in the original dimension $\mathbb{R}^d$ is

$$\sum_{x \in S_j} w_j(x)\|x - \nu^j\|^p.$$

Let $\nu^j$ denote the argmin of this objective in $\mathbb{R}^d$ and let $u^j$ denote the argmin for this same objective but in the projected space $\mathbb{R}^m$, i.e.

$$u^j = \operatorname*{argmin}_{u \in \mathbb{R}^m} \sum_{x \in S_j} w_j(x)\|\pi x - u\|^p.$$

Then we have

$$\sum_{x \in S_j} w_j(x)\|x - \nu^j\|^p \leq \sum_{x \in S_j} w_j(x)\|x - \widetilde{\varphi}(u^j)\|^p \qquad (\nu^j \text{ is more optimal than } \widetilde{\varphi}(u))$$

$$= \sum_{x \in S_j} w_j(x)\|\widetilde{\varphi}(\pi x) - \widetilde{\varphi}(u^j)\|^p \qquad (\widetilde{\varphi} = \pi^{-1} \text{ for } x \in S_j)$$

$$\leq \sum_{x \in S_j} w_j(x)(1+\varepsilon)^p\|\pi x - u^j\|^p \qquad (\widetilde{\varphi} \text{ is } (1+\varepsilon)\text{-Lipschitz})$$

$$= (1+\varepsilon)^p \sum_{x \in S_j} w_j(x)\|\pi x - u^j\|^p.$$

Summing over all $S_j$ finally leads to $\text{cost}(S) \leq (1+\varepsilon)^p\text{cost}(\pi S)$. A similar reasoning also gives $\text{cost}(\pi S) \leq (1+\varepsilon)^p\text{cost}(S)$ and combining these two statements and adjusting $\varepsilon$ proves the theorem. $\qquad \square$

# B Proofs for Section 4

In this section, we give the missing proofs from Section 4. Our main goal will be to prove Theorem 4.1; we also describe a "faster" dimension reduction map at the end of the section. To prove Theorem 4.1, we will actually first prove a version of the theorem with a slightly rescaled value of $\varepsilon$ (Theorem B.13). Theorem 4.1 follows immediately by "undoing" the rescaling.

We adapt this analysis to the Wasserstein barycenter problem by handling four additional issues: (i) the input points are weighted since they come from probability distributions; (ii) input points may be assigned to multiple support points in the barycenter; (iii) each barycenter point is constrained to receive a specific amount of mass under optimal transport; and (iv) the distorted points must not contribute large error to the cost induced by the Wasserstein barycenter. Issues (i) and (ii) are problematic because previous structural results for the distortion graph do not rule out a large weighted fraction of the distances being distorted. Furthermore, issues (iii) and (iv) are problematic because we cannot isolate each point in a probability distribution to a specific barycenter. We again consider a hypothetical distortion graph on the $k \cdot \text{poly}(n)$ points in $\mathbb{R}^d$ with nonzero support in the $k$ distributions and connect an edge between each pair of points if their pairwise distance is distorted by the random projection map $\pi$ by at least a $(1+\varepsilon)$-factor. To resolve issue (1), we give a combinatorial

argument that shows that the distortion graph for $\pi$ is everywhere sparse for a weighted notion of sparsity. To resolve issues (2) and (3), we define a mapping for each point in a probability distribution that partitions its mass among the barycenters. Using the everywhere-sparse distortion graph, we show a robust 1-point extension theorem that the pairwise distances from the barycenter to a large *weighted* fraction of the points is preserved. Finally to resolve issue (4), we show that the remaining weighted fraction of points incurs a cost that is at most $\varepsilon$-fraction of the optimal cost induced by the Wasserstein barycenter.

The structure of the proof is as follows. To prove that the cost of any solution (in the sense of Definition 2.1) is preserved, we first show that the cost of the flow from a weighted cluster of points in the $\mu_i$s to one particular support point in the barycenter is preserved (Theorem B.12). This in turn rests on the fact that weighted cluster costs are preserved when only a small weighted fraction of the cluster distances are distorted (Theorem B.1).

In summary, the overall proof structure is

$$\text{Theorem B.1} \implies \text{Theorem B.12} \implies \text{Theorem B.13} \implies \text{Theorem 4.1}.$$

We begin by proving Theorem B.1, which should be considered the weighted analog to Theorem 3.3 in [MMR19].

**Theorem B.1.** *Let $X \subset \mathbb{R}^d$ be a finite set of weighted points and the map $\phi : X \to \mathbb{R}^m$ have a distortion graph $G$ for $X$ that is $\alpha$-sparse (with respect to the weight of $X$), with $\alpha \leq 1/10^{p+1}$. Then for every $p \geq 1$ and $D = (1+\varepsilon)^p(1 + 3^{p+2} + \alpha^{1/(p+1)})$, we have*

$$\frac{1}{D}\mathrm{cost}_p(X) \leq \mathrm{cost}_p(\phi(X)) \leq D\mathrm{cost}_p(X)$$

*where $\mathrm{cost}_p$ is the cost of solving a clustering on $X$ with only $1$ center under the cost function $\|\cdot\|_2^p$.*

Unfortunately, the results of [MMR19] do not immediately imply the corresponding sparsity results for weighted graphs. For example, a vertex that has edges to a small fraction of its neighbors may still have an edge to a large weighted fraction of its neighbors. Thus we show the weighted analogs of the structural results from [MMR19]. The following lemma is analogous to Lemma 4.1 in [MMR19], extending the properties to handle weighted sets $X$.

**Lemma B.2.** *Let $X$ be a finite set and $V \subset X$ be a random subset of $X$. Let $\alpha \in (0, 1/2)$ and suppose that $\mathbf{Pr}\left[x \in V\right] \geq 2\alpha$ for each $x \in X$. Then there exist a random set $R \subset V$ and a deterministic measure $\mu$ on $X$ such that*

1. $\mu(x) \geq \frac{w(x)}{w(V \setminus R)}$ *for every $x \in V \setminus R$*

2. $\mathbf{Pr}\left[x \in R\right] \leq 2\alpha$ *for every $x \in X$*

3. $\mu(X) = \sum_{x \in X} \mu(x) \leq \frac{\mathbf{Pr}[V \neq \emptyset]}{\alpha^2}$

*Proof.* Since $X$ is a finite set, we truncate (or discretize) the weights of the elements in $X$ and without loss of generality suppose that there exists a sufficiently large integer $N > 0$ such that for each $x \in X$, there exists some integer $i \leq N$ such that $w(x) = \frac{i}{N}$. We then prove our claim by induction on the weight of the set $X$. If $w(X) = 0$ so that $X$ is empty, then the claim trivially holds. Now we suppose that $w(X) = \frac{k}{N}$ and the statement holds for all sets $X'$ with weight $w(X') = \frac{k'}{N}$, where $k' < k$ are non-negative integers; we show the statement holds for $X$.

Let $\ell = \alpha \cdot w(X)$ and define a deterministic set $X'$ and a random subset $V' \subset X'$ by:

$$X' = \{x : \mathbf{Pr}\left[x \in V \text{ and } w(V) < \ell\right] \geq 2\alpha\}$$

$$V' = \begin{cases} V \cap X', & \text{if } w(V) < \ell \\ \emptyset, & \text{otherwise} \end{cases}$$

We first show that there exists an $x_0 \in X$ such that $\mathbf{Pr}\left[x_0 \in V \text{ and } w(V) < \ell\right] \leq \alpha$, which implies that $x_0 \notin X'$ and thus $w(X') < w(X)$. We show that the average value of $\mathbf{Pr}\left[x \in V \text{ and } w(V) < \ell\right]$

for $x \in X$ is at most $\alpha$, which implies the existence of such an $x_0$. Since $w(V) \cdot \mathbb{1}\{w(V) < \ell\}$ is always at most $\ell$, then

$$\frac{1}{w(X)} \sum_{x \in X} \mathbf{Pr}\left[x \in V \text{ and } w(V) < \ell\right] = \frac{1}{w(X)} \sum_{x \in X} \mathbb{E}\left[\mathbb{1}\{x \in V \text{ and } w(V) < \ell\}\right]$$

$$= \frac{1}{w(X)} \mathbb{E}\left[\sum_{x \in X} \mathbb{1}\{x \in V \text{ and } w(V) < \ell\}\right] = \frac{1}{w(X)} \mathbb{E}\left[w(V) \cdot \mathbb{1}\{w(V) < \ell\}\right] \leq \frac{\ell}{w(X)} = \alpha.$$

Because $w(X') < w(X)$ and $\mathbf{Pr}\left[x \in V'\right] = \mathbf{Pr}\left[x \in V \text{ and } w(V) < \ell\right] \geq 2\alpha$ for each $x \in X'$ by definition of $X'$, then we apply the inductive hypothesis to $X'$ and $V'$. Hence, there exist a random set $R' \subset X'$ and a measure $\mu'$ on $X'$ such that the above claims 1-3 hold for $X'$ and $V'$. We then define a measure $\mu$ on $X$ and random subset $R \subset V$ by:

$$\mu(x) = \begin{cases} \mu'(x) + \frac{w(x)}{\ell}, & \text{if } x \in X' \\ \frac{w(x)}{\ell}, & \text{otherwise.} \end{cases}$$

$$R = \begin{cases} R' \cup (V \setminus X'), & \text{if } w(V) < \ell \\ R', & \text{otherwise.} \end{cases}$$

We claim that $R$ and $\mu$ satisfy the desired properties.

**Property 1:** $\mu(x) \geq \frac{w(x)}{w(V \setminus R)}$ **for each** $x \in V \setminus R$. Let $x \in V \setminus R$.
We have three possible cases. (1) If $x \in X'$ and $w(V) < \ell$, then $V \setminus R = V' \setminus R'$ by the definition of $R$. Hence, $\mu(x) > \mu'(x) \geq \frac{w(x)}{w(V' \setminus R)} = \frac{w(x)}{w(V \setminus R)}$ by the inductive hypothesis. (2) If $x \in X'$ and $w(V) \geq \ell$, then $\mu(x) \geq \frac{w(x)}{\ell} \geq \frac{w(x)}{w(V)}$. Since $w(V) \geq \ell$, we also have $V' = \emptyset$, so by the definition of $R$, we have $R = R' \subset V' = \emptyset$. Thus, $\frac{w(x)}{w(V)} = \frac{w(x)}{w(V \setminus R)}$ so that $\mu(x) \geq \frac{w(x)}{w(V \setminus R)}$. (3) If $x \notin X'$, then $x \in V \setminus X' \subset R' \cup (V \setminus X')$. Since $x \in V \setminus R$, then $x \notin R$. Thus, $R \neq R' \cup (V \setminus X')$. By the definitions of $\mu$ and $R$, we have that $w(V) \geq \ell$ and $\mu(x) = \frac{w(x)}{\ell}$, so that $\mu(x) \geq \frac{w(x)}{w(V)}$. Since $w(V) \geq \ell$, then $\mu(x) \geq \frac{w(x)}{w(V)} = \frac{w(x)}{w(V \setminus R)}$.

**Property 2:** $\mathbf{Pr}\left[x \in R\right] \leq 2\alpha$. If $x \in X'$, then by the inductive hypothesis, $\mathbf{Pr}\left[x \in R\right] = \mathbf{Pr}\left[x \in R'\right] \leq 2\alpha$. If $x \notin X'$, then by the definitions of $R$ and $X'$ respectively, we have that $\mathbf{Pr}\left[x \in R\right] = \mathbf{Pr}\left[x \in V \text{ and } w(V) < \ell\right] \leq 2\alpha$.

**Property 3:** $\mu(X) = \sum_{x \in X} \mu(x) \leq \mathbf{Pr}\left[V \neq \emptyset\right]/\alpha^2$. By the inductive hypothesis, $\mu'(X') \leq \mathbf{Pr}\left[V' \neq \emptyset\right]/\alpha^2$. Therefore, $\mu(X) = \mu'(X') + \frac{w(X)}{\ell} \leq \frac{\mathbf{Pr}\left[V' \neq \emptyset\right]}{\alpha^2} + \frac{1}{\alpha}$. Note that $\mathbf{Pr}\left[V \neq \emptyset\right] - \mathbf{Pr}\left[V' \neq \emptyset\right] \geq \alpha$ implies $\frac{\mathbf{Pr}\left[V' \neq \emptyset\right]}{\alpha^2} + \frac{1}{\alpha} \leq \frac{\mathbf{Pr}\left[V \neq \emptyset\right]}{\alpha^2}$, which would imply the desired claim; thus it suffices to prove $\mathbf{Pr}\left[V \neq \emptyset\right] - \mathbf{Pr}\left[V' \neq \emptyset\right] \geq \alpha$.

Observe that if $w(V) \geq \ell$, then $V \neq \emptyset$ but $V' = \emptyset$. Since $V' \subset V$, then

$$\mathbf{Pr}\left[V \neq \emptyset\right] - \mathbf{Pr}\left[V' \neq \emptyset\right] = \mathbf{Pr}\left[V \neq \emptyset \text{ and } V' = \emptyset\right] \geq \mathbf{Pr}\left[w(V) \geq \ell\right].$$

Recall that we previously showed the existence of an $x_0 \in X$ with $\mathbf{Pr}\left[x_0 \in V \text{ and } w(V) < \ell\right] \leq \alpha$. On the other hand, $\mathbf{Pr}\left[x \in V\right] \geq 2\alpha$ for all $x \in X$. Thus,

$$\mathbf{Pr}\left[w(V) \geq \ell\right] \geq \mathbf{Pr}\left[x_0 \in V \text{ and } w(V) \geq \ell\right] \geq \mathbf{Pr}\left[x_0 \in V\right] - \mathbf{Pr}\left[x_0 \in V \text{ and } w(V) < \ell\right] \geq \alpha.$$

Hence, we have shown that all three properties are satisfied by $R$ and $\mu$, which completes the induction for $X$. $\qquad\square$

The following claim is analogous to Corollary 4.2 in [MMR19], again extending the properties to handle weighted sets $X$.

**Corollary B.3.** *Let $X$ be a finite set, $V \subset X$ be a random subset of $X$, and $\alpha \in (0, 1/2)$. Then there exist a random set $R \subset V$ and a measure $\mu$ on $X$ such that*

1. $\mu(x) \geq \frac{w(x)}{w(V \setminus R)}$ for every $x \in V \setminus R$

2. $\mathbf{Pr}\left[x \in R\right] \leq 2\alpha$ for every $x \in X$

3. $\mu(X) = \sum_{x \in X} \mu(x) \leq \frac{1}{\alpha^2}$

*Proof.* Let $X' = \{x : \mathbf{Pr}\left[x \in V\right] \geq 2\alpha$. By applying Lemma B.2 to $X'$ and $V' = V \cap X'$ and set

$$R = R' \cup (V \setminus X'),$$

$$\mu(x) = \begin{cases} \mu'(x), & \text{if } x \in X' \\ 0, & \text{otherwise} \end{cases}.$$

$\square$

We also have the following analog to Observation 4.3 in [MMR19].

**Observation B.4.** *Let $R$ be defined as in Corollary B.3 and $V_0 = V \setminus R$. Then for every $S \subset V_0$, we have $w(S) \leq \mu(S) \cdot w(V_0)$.*

*Proof.* By Corollary B.3, we have $\mu(x) \geq w(x)/w(V_0)$ for every $x \in S$. Thus, $\mu(S) \geq w(S)/w(V_0)$, so $w(S) \leq \mu(S) \cdot w(V_0)$. $\square$

The following is analogous to Theorem 3.2 in [MMR19].

**Theorem B.5** (Theorem 3.2 in [MMR19])**.** *Consider a finite set $X$ and a random graph $H = (V, E)$, where $V$ is a random subset of $X$ and $E$ is a random set of edges between vertices in $V$ (there are no independence assumptions or any other implicit assumptions about the distribution of $V$ and $E$). Let $\alpha \in (0, 1/2)$. Assume that $\mathbf{Pr}\left[(x, y) \in E\right] \leq \delta \leq \alpha$ for every $x, y \in X$. Then there exists a random subset $V' \subset V$ such that*

- *$H[V']$ is $\alpha$-everywhere sparse,*

- *$\mathbf{Pr}\left[u \in V \setminus V'\right] \leq 600\delta/\alpha^6$ for all $u \in X$.*

*Proof.* Let $\beta = \alpha/(1 + \alpha)$ and $\alpha' = \alpha/3$. Applying Corollary B.3 with $\alpha'$, we get a deterministic measure $\mu$ on $V$ and a random set $R \subset V$. Consider the canonical product measure

$$\mu^{\otimes 2}((x, y)) = \mu(x)\mu(y)$$

for all $x, y \in X$. We define $V'$ according to the measure $\mu^{\otimes 2}$.

**Case 1**: $\mu^{\otimes 2}(E) \geq \beta^2$. In this case, we let $V' = \emptyset$.

**Case 2**: $\mu^{\otimes 2}(E) < \beta^2$. In this case, define $V_0 = V \setminus R$. We say that $x \in X$ is bad if $\mu(\{y \in V_0 : (x, y) \in E\}) \geq \beta$. Let $B$ denote the set of bad vertices and define

$$V' = V \setminus (R \cup B) = V_0 \setminus B.$$

Our goal is to verify that in both of the above cases, both of the desired properties of Theorem B.5 hold. First we handle the easier Case 1. There, the graph $H[V']$ is empty so the conclusion trivially holds. For the second condition, note that

$$\mathbf{Pr}\left[\mu^{\otimes 2}(E) \geq \beta^2\right] \leq \frac{\mathbb{E}\left[\mu^{\otimes 2}(E)\right]}{\beta^2} \leq \frac{\delta M^2}{\beta^2}$$

where $M = \mu(X)$ and we have used the fact that $\mathbf{Pr}\left[(x, y) \in E\right] \leq \delta$. Then from our choice of $\beta$ and $\alpha'$ (fill in details in a bit), the above probability is at most $C\alpha$.

We now verify the second case. First, we check that $H[V']$ is $\alpha$-everywhere sparse. This is equivalent to checking that the weighted degree of every vertex $x$ in $H[V']$ is at most $\alpha$ fraction of the total weight. That is, we need to check:

$$w(\{y \in V' : (x, y) \in E\}) \leq \alpha w(V').$$

Analogous to the proof of Theorem 3.2 in [MMR19], we have $\mu^{\otimes 2}(E) \geq \beta\mu(B)$ and therefore, $\mu(V_0 \cap B) \leq \mu(B) \leq \beta$ since we are in Case 2. Now for $x \in V'$, we similarly have $\mu(y \in V' : (x, y) \in V') \leq \beta$. Combining the above findings with Observation B.4, we conclude the following two statements:

- $w(\{y \in V' : (x, y) \in E\}) \leq \beta w(V_0)$,

- $w(V_0 \cap B) \leq \beta w(V_0)$.

From the second relation, we have $w(V') = w(V_0 \setminus B) \geq (1 - \beta)w(V_0)$ and using the first relation, we can conclude that

$$w(\{y \in V' : (x, y) \in E\}) \leq \frac{\beta}{1 - \beta}w(V').$$

Finally, the same probability bound as in the end of the proof of Theorem 3.2 in [MMR19] allows us to say that in Case 2, the probability of $x \in B$ is at most $\delta M/\beta$. Finally, combining all the probabilities from Case 1 and Case 2, we can conclude identically as in Theorem 3.2 in [MMR19] that $\mathbf{Pr}\left[u \in V\ V'\right] \leq \delta M^2/\beta^2 + 2\alpha' + \delta M/\beta \leq 600\delta/\alpha^6$. $\qquad\square$

We now state an analogous version of Theorem 5.2 from [MMR19] that is suitable for our purposes.

**Theorem B.6** (Robust Kirszbraun Theorem). *Consider two finite (multi) sets of points $X \subset \mathbb{R}^d$ and $Y \subset \mathbb{R}^m$ and a map $\varphi : X \to Y$. Let $G = (X, E)$ be the distance expansion graph for $\varphi$ with respect to the Euclidean distance with vertex weights given by $w : X \to \mathbb{R}^{\geq 0}$. Suppose that $G$ is $\alpha$-everywhere sparse according to Definition B.10. Then for every $u \in \mathbb{R}^d$ and $\varepsilon > 0$, there exists $v \in \mathbb{R}^m$ and $X' \subset X$ such that for all $x \in X \setminus X'$,*

$$\|\phi(x) - v\| \leq (1 + \varepsilon)\|x - u\|,$$

*and $w(X') \leq \alpha'(\varepsilon)w(X)$ where $\alpha'(\varepsilon) = 2(1 + \varepsilon)^2\alpha/\varepsilon$.*

The proof of Theorem 5.2 from [MMR19] carries over in a straightforward fashion to the proof of Theorem B.6 above. In particular, we just outline the small changes that need to occur to carry the proof over.

*Proof Sketch.* In [MMR19], the following polytope is defined:

$$\Lambda_\eta = \{\lambda \in \mathbb{R}^X : \sum_{x \in X} \lambda_x = 1; 0 \leq \lambda_{x'} \leq \eta \text{ for all } x' \in X\}$$

for $\eta = (\alpha'(\varepsilon)n)^{-1}$. For us, we define a slightly modified polytope which includes the weights of elements of $X$:

$$\Lambda_\eta = \{\lambda \in \mathbb{R}^X : \sum_{x \in X} \lambda_x = 1; 0 \leq \lambda_{x'} \leq w(x')\eta \text{ for all } x' \in X\}$$

where $\eta = (\alpha'(\varepsilon)w(X))^{-1}$. Then for every $\lambda \in \Lambda_\eta$, $u' \in \mathbb{R}^{d'}$, and $v' \in R^{d''}$, we similarly let

$$f(X, \lambda, u') = \sum_{x \in X} \lambda_x\|u' - x\|^2 \text{ and } f(\phi(X), \lambda, v') = \sum_{x \in X} \lambda_x\|v' - \phi(x)\|^2.$$

Since $\Lambda_\eta$ is still a convex polytope, we recover the statement

$$\max_{v' \in \mathbb{R}^{d''}} \min_{\lambda \in \Lambda_\eta} F(v', \lambda) \geq 0 \tag{5}$$

where $F(v', \lambda) = (1 + \varepsilon)^2 f(X, \lambda, u) - f(\varphi(X), \lambda, v')$. Now to finish the rest of the proof, let $v$ be the point that maximizes the functional $\min_{\lambda \in \Lambda_\eta} F(v, \lambda)$. By (5), we know that $F(v, \lambda) \geq 0$ for all $\lambda \in \Lambda_\eta$. Now consider the set

$$S = \{x \in X : \|\varphi(x) - v\| \geq (1 + \varepsilon)\|x - u\|\}.$$

If $S = \emptyset$, we are done so otherwise, define $\lambda^*$ as

$$\lambda_x^* = \begin{cases} \frac{w(x)}{w(S)} & \text{if } x \in S, \\ 0 & \text{otherwise} . \end{cases}$$

By the definition of $S$, we have $(1+\varepsilon)\|x-u\|^2 - \|\varphi(x)-v\|^2 < 0$ which implies

$$F(v,\lambda^*) = \frac{1}{w(S)}\sum_{x\in S} w(x)\left((1+\varepsilon)\|x-u\|^2 - \|\varphi(x)-v\|^2\right) < 0$$

and thus, $\lambda^* \notin \Lambda_\eta$. Therefore, $1/w(S) > \eta$ and $w(S) < 1/\eta = \alpha'(\varepsilon)w(X)$. This finishes the proof of Theorem B.6. $\qquad\square$

We also need the following analogous version of Lemma 5.3 in [MMR19]. The proof differs in that we have to consider a careful weighting scheme whereas in [MMR19] it was more straightforward. We first require the following property:

**Lemma B.7** (Lemma A.1 in [MMR19])**.** *Let $x$ and $y_1,\ldots,y_r$ be non-negative real numbers, and $\varepsilon > 0$, $p \geq 1$. Then*

$$\left(x + \sum_{i=1}^r y_i\right)^p \leq (1+\varepsilon)^{p-1}x^p + \left(\frac{(1+\varepsilon)r}{\varepsilon}\right)^{p-1}\sum_{i=1}^r y_i^p.$$

**Lemma B.8** (Lemma 5.3 in [MMR19])**.** *Consider two finite multisets of points $X \subset \mathbb{R}^d$ and $Y \subset \mathbb{R}^m$ of the same size and a one-to-one map $\varphi : X \to Y$. Let $G = (X,E)$ be the distance expansion graph for $\varphi$ with respect to the Euclidean distance with a weight function $w : X \to \mathbb{R}^{\geq 0}$. Suppose that $G$ is $\alpha$-everywhere sparse with $\alpha \leq 1/10^{p+1}$. Then, for every $p \geq 1$, we have the following inequality on the cost of the clusters $X$ and $Y$ (with the same weights as $w$)*

$$\mathrm{cost}_p(Y) \leq (1 + 3^{p+2}\alpha^{1/(p+1)})\mathrm{cost}_p(X).$$

*Proof.* Let $\varepsilon = \alpha^{1/(p+1)}$ and let $u^*$ be the optimal center for the cluster $X$. By Theorem B.6, there exists a set $\widetilde{X} \subset X$ and a point $v^* \in \mathbb{R}^m$ such that for $x \in \widetilde{X}$,

$$\|\varphi(x) - v^*\| \leq (1+\varepsilon)\|x - u^*\|$$

where $w(X \setminus \widetilde{X}) \leq \alpha' w(X)$ with $\alpha' \leq 2(1+\varepsilon)^2\alpha/\varepsilon$. By definition, it follows that

$$
\begin{aligned}
\mathrm{cost}_p(Y) &\leq \sum_{y\in Y} w(y)\|y - v^*\|^p \\
&= \sum_{x\in X} w(x)\|\varphi(x) - v^*\|^p \quad (x \text{ and } \varphi(x) = y \text{ have the same weight}) \\
&= \sum_{x\in\widetilde{X}} w(x)\|\varphi(x) - v^*\|^p + \sum_{x\notin\widetilde{X}} w(x)\|\varphi(x) - v^*\|^p \\
&\leq (1+\varepsilon)^p \sum_{x\in\widetilde{X}} w(x)\|x - u^*\|^p + \sum_{x\notin\widetilde{X}} w(x)\|\varphi(x) - v^*\|^p.
\end{aligned}
$$

We now try to bound $\|\varphi(x) - v^*\|^p$ for $x \notin \widetilde{X}$. We will bound this quantity using a slightly stronger claim that applies for all $x \in X$. Indeed, fix an arbitrary $x$ and consider the set $I_x$ of its non neighbors in the distance expansion graph $G$. Note that $w(I_x) \geq (1-\alpha)w(X)$ and thus, $w(I_x \cap \widetilde{X}) \geq (1-\alpha-\alpha')w(X) > 0$ if the total weight $w(X)$ is positive. Consider an arbitrary $x' \in I_x \cap \widetilde{X}$. Then it follows that

$$
\begin{aligned}
\|\varphi(x) - v^*\| &\leq \|\varphi(x) - \varphi(x')\| + \|\varphi(x') - v^*\| \\
&\leq (1+\varepsilon)\|x - x'\| + (1+\varepsilon)\|x' - u^*\| \quad (\text{using the fact that } x' \in I_x \cap \widetilde{X}) \\
&= (1+\varepsilon)\|x - u^*\| + (2+2\varepsilon)\|x' - u^*\|.
\end{aligned}
$$

Applying Lemma B.7, we have that

$$\|\varphi(x) - v^*\| \leq (1+\varepsilon)^p\|x - u^*\|^p + \frac{3^p}{\varepsilon^{p-1}}\|x' - u^*\|^p \tag{6}$$

for $\varepsilon$ sufficiently small. We now average equation (6) over all possible $x'$. This gives us

$$\|\varphi(x) - v^*\| \le (1+\varepsilon)^p \|x - u^*\|^p + \frac{3^p}{\varepsilon^{p-1}} \cdot \frac{1}{w(I_x \cap \widetilde{X})} \sum_{x' \in I_x \cap \widetilde{X}} w(x') \|x' - u^*\|^p$$

$$\le (1+\varepsilon)^p \|x - u^*\|^p + \frac{3^p}{\varepsilon^{p-1}} \cdot \frac{1}{\alpha'' w(X)} \sum_{x' \in I_x \cap \widetilde{X}} w(x') \|x' - u^*\|^p$$

where $\alpha'' = (1 - \alpha - \alpha')$. Therefore,

$$\sum_{x \notin \widetilde{X}} w(x) \|\varphi(x) - v^*\|^p \le (1+\varepsilon)^p \sum_{x \notin \widetilde{X}} w(x) \|x - u^*\|^p$$

$$+ \frac{3^p}{\varepsilon^{p-1}} \cdot \frac{1}{\alpha'' w(X)} \sum_{\substack{x \notin \widetilde{X} \\ x' \in I_x \cap \widetilde{X}}} w(x) w(x') \|x' - u^*\|^p.$$

Focusing on the second term, we have

$$\frac{1}{\alpha'' w(X)} \sum_{\substack{x \notin \widetilde{X} \\ x' \in I_x \cap \widetilde{X}}} w(x) w(x') \|x' - u^*\|^p \le \frac{1}{\alpha'' w(X)} \sum_{x \notin \widetilde{X}} w(x) \sum_{x' \in \widetilde{X}} \|x' - u^*\|^p$$

$$\le \frac{1}{\alpha'' w(X)} \sum_{x \notin \widetilde{X}} w(x) \operatorname{cost}_p(X)$$

$$\le \operatorname{cost}_p(X) \frac{w(X \setminus \widetilde{X})}{\alpha'' w(X)}$$

$$\le 2\alpha' \operatorname{cost}_p(X)$$

using the fact that $\alpha'' \ge 1/2$. Putting everything together gives us

$$\operatorname{cost}_p(Y) \le (1+\varepsilon)^p \sum_{x \in X} w(x) \|x - u^*\|^p + \frac{3^p}{\varepsilon^{p-1}} \cdot 2\alpha' \operatorname{cost}_p(X)$$

$$\le (1 + 3^{p+2}\alpha') \operatorname{cost}_p(X)$$

for $\varepsilon$ sufficiently small. $\qquad\square$

Finally, with the above lemmas in hand, we are ready to prove Theorem B.1.

*Proof of Theorem B.1.* Given Lemma B.8, the proof follows identically as the proof of Theorem 3.2 in [MMR19] by applying Lemma B.8 to the maps $(1+\varepsilon)\varphi$ and $(1+\varepsilon)\varphi^{-1}$. $\qquad\square$

Next we prove Theorem B.12, which shows that the cost of each "cluster" in the barycenters problem (i.e. the cost of the weighted flow from points in the input distributions to *one* of the support points in the barycenter) is preserved. We will make use of *distortion graphs*, which quantify the level of distortion of pairwise distances resulting from a dimensionality reduction map.

**Definition B.9.** *Let $\pi : \mathbb{R}^d \to \mathbb{R}^m$ and $X$ be a set of points in $\mathbb{R}^d$. A distortion graph $G$ with vertex set $X$ is a graph where two points $u, v \in X$ are joined by an edge if the distance between $u$ and $v$ is distorted by a factor at least $1 + \varepsilon$ by $\pi$.*

We define the following concept of an everywhere sparse graph to be a generalization to weighted graphs of the concept introduced by [MMR19].

**Definition B.10.** *Let $G = (V, E)$ be a graph with vertex weights given by $w : V \to \mathbb{R}^{\ge 0}$. Let $N(u)$ denote the neighborhood of a vertex $u$. $G$ is $\alpha$-everywhere sparse if*

$$\sum_{v \in N(u)} w(v) \le \alpha w(V) = \alpha \sum_{v \in V} w(v)$$

*for all $u \in V$.*

Finally, we impose some additional requirements on the dimensionality reduction map. These essentially say that, even when a pair of points are distorted by the reduction map, the distortion is not too large in expectation.

**Definition B.11.** *For $\varepsilon > 0$, $\delta \in (0,1)$, a random map $\pi : \mathbb{R}^d \to \mathbb{R}^m$ is an $(\varepsilon, \delta)$-dimension reduction if*

$$\frac{1}{1+\varepsilon}\|x - y\| \leq \|\pi(x) - \pi(y)\| \leq (1+\varepsilon)\|x - y\|,$$

*with probability at least $1 - \delta$ for every $x, y \in \mathbb{R}^d$. For $p \geq 1$, $\pi$ is an $(\varepsilon, \delta, \alpha)$-dimension reduction if it additionally satisfies*

$$\mathbb{E}\left[\mathbb{1}\{\|\pi(x) - \pi(y)\| > (1+\varepsilon)\|x - y\|\}\left(\frac{\|\pi(x) - \pi(y)\|^p}{\|x - y\|^p} - (1+\varepsilon)^p\right)\right] \leq \alpha.$$

*We say $\pi$ is a standard dimension reduction if the parameters $(\varepsilon, \delta, \alpha)$ permit $\delta \leq \exp(-C\varepsilon^2 d)$ and $\alpha \leq \exp(-C\varepsilon^2 d)$ for $d \geq C'p/\varepsilon^2$ for some absolute constants $C, C' > 0$.*

In the theorem below, we just consider one point $\nu^j$ in the support of the barycenter and corresponding points (and weights) $S_j, w_j(\cdot)$ assigned to $\nu^j$. We show that a random $(\varepsilon, \delta, \alpha)$-standard dimensionality reduction map roughly preserves the cost of the assignment. The proof is similar to Theorem 3.4 in [MMR19], but we instead use a weighted version of the distortion graphs.

**Theorem B.12.** *Let $\mu_1, \ldots, \mu_k$ be an instance of the Wasserstein barycenter problem with the $L_p$ objective. Define $X = \bigcup_{i=1}^k \operatorname{supp}(\mu_i)$. Let $\pi$ be a random $(\varepsilon, \delta, \alpha)$-standard dimensionality reduction map and let $\nu_* = \sum_{j=1}^n b_j \delta(c_j^*)$ be the Wasserstein barycenter. Furthermore, let $\mathcal{C}^* = (C_1^*, \ldots, C_n^*)$ denote the solution to the minimum flow problem for the $\mu_i$s to $\nu^*$ with corresponding weight functions $w_j^*(\cdot)$ (see Definition 2.1).*

*Let $\mathcal{C} = (C_1, \ldots, C_n)$ with corresponding weight functions $w_j(\cdot)$ be any solution (possibly random that depends on $\pi$) to the Wasserstein barycenter problem in the sense of Definition 2.1. Let $C = C_j$ be any fixed cluster in $\mathcal{C}$, and further suppose that $\alpha \leq 1/10^{p+1}$ and $\delta \leq \min(\alpha^7/600, \alpha/n)$. Then with probability at least $1 - \eta - \binom{n}{2}\delta$,*

$$\operatorname{cost}_p(\pi(C)) \leq A(\operatorname{cost}_p(C) + c_{\eta\varepsilon\alpha}\operatorname{cost}_p(\mathcal{C}^*))$$
$$\operatorname{cost}_p(C) \leq A(\operatorname{cost}_p(\pi(C)) + c_{\eta\varepsilon\alpha}\operatorname{cost}_p(\mathcal{C}^*)),$$

*where $A = (1+\varepsilon)^{3p-2}(1 + 3^{p+2}\alpha^{1/(p+1)})$ and $c_{\eta\varepsilon\alpha} = \frac{5(1+\varepsilon)^p\alpha}{\eta\varepsilon^{p-1}}$.*

*Proof.* We use a similar outline to the proof of Theorem 3.4 in [MMR19]. Fix $C = C_j$ and let $w_C(\cdot) \equiv w_j(\cdot)$. Let $\mathcal{E}$ be the event that all distances between the points $c_i^*$ are preserved within a $(1+\varepsilon)$-approximation, so that $\mathbf{Pr}[\mathcal{E}] \geq 1 - \binom{n}{2}\delta$ under a random JL projection (or other random standard dimensionality reduction projection). We thus condition the remainder of the proof on the event $\mathcal{E}$. Let $C^\circ \subset C$ be the subset of $C$ whose distances to each center $c_i^*$ are preserved within a $(1+\varepsilon)$-approximation, so that

$$C^\circ = \{x \in C : \pi \text{ preserves the distance between } x \text{ and each } c_i^* \text{ within a factor of } (1+\varepsilon)\}.$$

Note that for a particular $x \in X$, we have that $\mathbf{Pr}[x \in C \setminus C^\circ] \leq n\delta$, since by a union bound, the probability that the distance between $x$ and some center $c_i^*$ is distorted by more than a $(1+\varepsilon)$-approximation is at most $n\delta$. Let $G[C^\circ]$ be the graph induced by $C^\circ$ on the distortion graph $G$. By Theorem B.5, there exists a set $C' \subset C^\circ$ such that $G[C']$ is $\alpha$-everywhere sparse and $\mathbf{Pr}[x \in C^\circ \setminus C'] \leq \alpha$. Thus,

$$\mathbf{Pr}[x \in C \setminus C'] \leq \mathbf{Pr}[x \in C \setminus C^\circ] + \mathbf{Pr}[x \in C^\circ \setminus C'] \leq n\delta + \alpha \leq 2\alpha.$$

We define $g(x)$ to be the identity mapping if $x \in C'$ (so that $x$ is a vertex of the $\alpha$-everywhere sparse graph) and otherwise, we define $g(x)$ to be the weighted multiset of the assignment of $x$ to each point in the support of the optimal Wasserstein barycenter $\mathcal{C}^*$:

$$g(x) = \begin{cases} (w_C(x), x), & \text{if } x \in C' \\ \{(w_C(x) \cdot r_i(x), c_i^*)\}_{i \in [n]}, & \text{if } x \notin C', \end{cases}$$

where $r_i(x)$ is the ratio of the weight of $x$ that is assigned to the point $c_i^*$. That is, $r_i(x) = w_i^*(x)/a(x)$, where recall $a(x) = $ weight of $x$ in $\mu_i$. Note that since $\sum_{i=1}^n w_i^*(x) = a(x)$ in order for $w_i^*(\cdot)$ to define a valid solution to the min flow problem, we have $\sum_{i=1}^n r_i(x) = 1$.

Let $\tilde{C} = g(C)$ be a multiset, so that every weighted point in $\tilde{C}$ is either assigned to a point in $C'$ or assigned to some point(s) $c_i^*$ in $\mathcal{C}^*$. Let $\tilde{c}$ be the optimal center for $\tilde{C}$. That is,

$$\tilde{c} = \operatorname*{argmin}_c \sum_{(w,y) \in \tilde{C}} w\|y - c\|^p.$$

Observe that since $G[C']$ is $\alpha$-everywhere sparse, then the map $\pi$ $(1 + \varepsilon)$-approximates the distances from every $x \in C'$ to (1) at least a $(1 - \alpha)$ weighted fraction of the points in $C'$ and (2) to all of the points in the barycenter $c_i^*$. Conditioning on the event $\mathcal{E}$ so that all pairwise distances between the points $c_i^*$ are approximated within a $(1 + \varepsilon)$ factor, then by Theorem B.1,

$$\frac{1}{D}\text{cost}_p(\tilde{C}) \le \text{cost}_p(\pi(\tilde{C})) \le D\text{cost}_p(\tilde{C}), \tag{7}$$

where $D = (1 + \varepsilon)^p(1 + 3^{p+2}\alpha^{1/(p+1)})$. Because $\tilde{c}$ is the optimal center for $\tilde{C}$, we further have

$$\text{cost}_p(C) \le \sum_{x \in C} w_C(x)\|x - \tilde{c}\|^p, \qquad \text{cost}_p(\tilde{C}) = \sum_{x \in C} \sum_{y \in g(x)} w\|y - \tilde{c}\|^p,$$

where we denote each ordered pair in $g(x)$ as $(w, y)$. We compare each term in the right hand side of the relationships for $\text{cost}_p(C)$ and $\text{cost}_p(\tilde{C})$. For $x \in C'$, we have that $g(x) = x$ and $w = w_C(x)$ so the contributions of each term in both summations are the same. For $x \notin C'$, the contributions are $w_C(x)\|x - \tilde{c}\|^p$ and $w_C(x)\sum_{i=1}^n r_i(x)\|c_i^* - \tilde{c}\|^p$ respectively. Hence,

$$\text{cost}_p(C) - (1+\varepsilon)^{p-1}\text{cost}_p(\tilde{C}) \le \sum_{x \in C \setminus C'}\left(w_C(x)\|x - \tilde{c}\|^p - (1 + \varepsilon)^{p-1}w_C(x)\sum_{i=1}^n r_i(x)\|c_i^* - \tilde{c}\|^p\right).$$

By triangle inequality, we have that $\|x - \tilde{c}\| \le \|x - c_i^*\| + \|c_i^* - \tilde{c}\|$ for each $i$. By Lemma B.7 with $r = 1$,

$$r_i(x)\|x - \tilde{c}\|^p \le (1 + \varepsilon)^{p-1}r_i(x)\|c_i^* - \tilde{c}\|^p + \left(\frac{1 + \varepsilon}{\varepsilon}\right)^{p-1}r_i(x)\|x - c_i^*\|^p \tag{8}$$

for each $i$. Recalling that $\sum_{i=1}^n r_i(x) = 1$ and summing over inequality (8), we have

$$\|x - \tilde{c}\|^p \le (1 + \varepsilon)^{p-1}\sum_{i=1}^n r_i(x)\|c_i^* - \tilde{c}\|^p + \left(\frac{1 + \varepsilon}{\varepsilon}\right)^{p-1}\sum_{i=1}^n r_i(x)\|x - c_i^*\|^p.$$

Thus,

$$\text{cost}_p(C) - (1 + \varepsilon)^{p-1}\text{cost}_p(\tilde{C}) \le \left(\frac{1 + \varepsilon}{\varepsilon}\right)^{p-1}\sum_{x \in C \setminus C'} w_C(x)\sum_{i=1}^n r_i(x)\|c_i^* - x\|^p.$$

By similar reasoning, we also have

$$\text{cost}_p(\tilde{C}) - (1 + \varepsilon)^{p-1}\text{cost}_p(C) \le \left(\frac{1 + \varepsilon}{\varepsilon}\right)^{p-1}\sum_{x \in C \setminus C'} w_C(x)\sum_{i=1}^n r_i(x)\|c_i^* - x\|^p \tag{9}$$

$$\text{cost}_p(\pi(C)) - (1 + \varepsilon)^{p-1}\text{cost}_p(\pi(\tilde{C})) \le \left(\frac{1 + \varepsilon}{\varepsilon}\right)^{p-1}\sum_{x \in C \setminus C'} w_C(x)\sum_{i=1}^n r_i(x)\|\pi(c_i^*) - \pi(x)\|^p \tag{10}$$

$$\text{cost}_p(\pi(\tilde{C})) - (1 + \varepsilon)^{p-1}\text{cost}_p(\pi(C)) \le \left(\frac{1 + \varepsilon}{\varepsilon}\right)^{p-1}\sum_{x \in C \setminus C'} w_C(x)\sum_{i=1}^n r_i(x)\|\pi(c_i^*) - \pi(x)\|^p. \tag{11}$$

Along with (7), we have that

$$\text{cost}_p(\pi(C)) = \Big[\text{cost}_p(\pi(C)) - (1+\varepsilon)^{p-1}\text{cost}_p(\pi(\tilde{C}))\Big] + (1+\varepsilon)^{p-1}\text{cost}_p(\pi(\tilde{C}))$$

$$\leq (10) + (1+\varepsilon)^{p-1}D\text{cost}_p(\tilde{C}) \qquad \text{(due to (7))}$$

$$= (10) + (1+\varepsilon)^{p-1}D\left[\Big(\text{cost}_p(\tilde{C}) - (1+\varepsilon)^{p-1}\text{cost}_p(C)\Big) + (1+\varepsilon)^{p-1}\text{cost}_p(C)\right]$$

$$\leq (10) + (1+\varepsilon)^{p-1}D\left[(9) + (1+\varepsilon)^{p-1}\text{cost}_p(C)\right].$$

Substituting the right-hand sides of (10) and (9) into the above and factoring out $A = (1+\varepsilon)^{2(p-1)}$, if we define

$$R_x = w_C(x)\sum_{i=1}^n \left(\underbrace{r_i(x)\|c_i^* - x\|^p}_{(A)} + \underbrace{r_i(x)\|\pi(c_i^*) - \pi(x)\|^p}_{(B)}\right) \tag{12}$$

we see that

$$\text{cost}_p(\pi(C)) \leq A\left(\text{cost}_p(C) + \varepsilon^{1-p}\sum_{c\in C\backslash C'} R_x\right). \tag{13}$$

A similar calculation yields

$$\text{cost}_p(C) \leq A\left(\text{cost}_p(\pi(C)) + \varepsilon^{1-p}\sum_{c\in C\backslash C'} R_x\right), \tag{14}$$

and inequalities (13) and (14) simultaneously hold with probability at least $1 - \binom{n}{2}\delta$.

Finally, we prove that $\varepsilon^{1-p}\sum_{x\in C\backslash C'} \mathbb{1}\{\mathcal{E}\}R_x \leq c_{\eta\varepsilon\alpha}\text{cost}_p(\mathcal{C}^*)$ with probability at least $1 - \eta$, by first showing that $\mathbb{E}\left[\mathbb{1}\{\mathcal{E}\}\sum_{x\in C\backslash C'} R_x\right] \leq 5(1+\varepsilon)^p\alpha\text{cost}_p(\mathcal{C}^*)$ and then applying Markov's inequality. We will bound the sum of the (B) terms first. Observe that

$$\|\pi(c_i^*) - \pi(x)\|^p \leq (1+\varepsilon)^p\|c_i^* - x\|^p + \max(\|\pi(c_i^*) - \pi(x)\|^p - (1+\varepsilon)^p\|c_i^* - x\|^p, 0).$$

Furthermore, we have

$$\max(\|\pi(c_i^*) - \pi(x)\|^p - (1+\varepsilon)^p\|c_i^* - x\|^p, 0) =$$

$$\mathbb{1}\{\|\pi(c_i^*) - \pi(x)\| > (1+\varepsilon)\|c_i^* - x\|\}\left(\|\pi(c_i^*) - \pi(x)\|^p - (1+\varepsilon)^p\|c_i^* - x\|^p\right).$$

Thus by Definition B.11, we have that for every $x\in C$ and any $i\in[n]$,

$$\mathbb{E}\left[\max(\|\pi(c_i^*) - \pi(x)\|^p - (1+\varepsilon)^p\|c_i^* - x\|^p, 0)\right] \leq \alpha\|c_i^* - x\|^p.$$

Combining these two bounds, we see that

$$\mathbb{E}\left[\sum_{x\in C\backslash C'} w_C(x)\sum_{i=1}^n r_i(x)\|\pi(c_i^*) - \pi(x)\|^p\right] \leq (1+\varepsilon)^p\,\mathbb{E}\left[\sum_{x\in C\backslash C'} w_C(x)\sum_{i=1}^n r_i(x)\|c_i^* - x\|^p\right]$$

$$+ \mathbb{E}\left[\sum_{x\in C\backslash C'} w_C(x)\sum_{i=1}^n r_i(x)\max(\|\pi(c_i^*) - \pi(x)\|^p - (1+\varepsilon)^p\|c_i^* - x\|^p, 0)\right].$$

The second term in the RHS can be bounded by

$$\mathbb{E}\left[\sum_{x \in C \setminus C'} w_C(x) \sum_{i=1}^{n} r_i(x) \max(\|\pi(c_i^*) - \pi(x)\|^p - (1+\varepsilon)^p \|c_i^* - x\|^p, 0)\right]$$

$$\leq \sum_{x \in X} \sum_{i=1}^{n} w_C(x) r_i(x) \mathbb{E}\left[\max(\|\pi(c_i^*) - \pi(x)\|^p - (1+\varepsilon)^p \|c_i^* - x\|^p, 0)\right]$$

$$\leq \alpha \sum_{x \in X} \sum_{i=1}^{n} w_i^*(x) \|c_i^* - x\|^p$$

$$\leq \alpha \operatorname{cost}_p(\mathcal{C}^*),$$

where we have replaced the sum over $C \setminus C'$ by a larger sum over all $X$ and used the fact that $w_C(x) r_i(x) \leq a(x) r_i(x) = w_i^*(x)$ for the weight $w_i^*(x)$ that is assigned to the point $c_i^*$ in the actual barycenter. Therefore,

$$\mathbb{E}\left[\sum_{x \in C \setminus C'} w_C(x) \sum_{i=1}^{n} r_i(x) \|\pi(c_i^*) - \pi(x)\|^p\right] \leq (1+\varepsilon)^p \mathbb{E}\left[\sum_{x \in C \setminus C'} w_C(x) \sum_{i=1}^{n} r_i(x) \|c_i^* - x\|^p\right]$$

$$\tag{15}$$

$$+ \alpha \operatorname{cost}_p(\mathcal{C}^*).$$

By linearity of expectation, we also have

$$\mathbb{E}\left[\sum_{x \in C \setminus C'} w_C(x) \|x - c_i^*\|^p\right] = \mathbb{E}\left[\sum_{x \in X} \mathbb{1}\{x \in C \setminus C'\} w_C(x) \|x - c_i^*\|^p\right]$$

$$= \sum_{x \in X} \mathbf{Pr}\left[x \in C \setminus C'\right] w_C(x) \|x - c_i^*\|^p$$

$$\leq 2\alpha \sum_{x \in X} w_C(x) \|x - c_i^*\|^p.$$

It then follows that

$$\mathbb{E}\left[\sum_{x \in C \setminus C'} w_C(x) \sum_{i=1}^{n} r_i(x) \|x - c_i^*\|^p\right] = \sum_{i=1}^{n} r_i(x) \mathbb{E}\left[\sum_{x \in C \setminus C'} w_C(x) \|x - c_i^*\|^p\right]$$

$$\leq \sum_{i=1}^{n} r_i(x) \cdot 2\alpha \sum_{x \in X} w_C(x) \|x - c_i^*\|^p$$

$$\leq 2\alpha \sum_{i=1}^{n} \sum_{x \in X} w_i^*(x) \|x - c_i^*\|^p$$

$$= 2\alpha \operatorname{cost}_p(\mathcal{C}^*) \tag{16}$$

where we again use the fact that $w_C(x) r_i(x) \leq a(x) r_i(x) = w_i^*(x)$. Notice that the bound obtained by (16) suffices to bound the first term in (15) *and* the (A) terms from (12). Combining bounds (15) and (16) with the definition of $R_x$ in (12), we obtain

$$\mathbb{E}\left[\mathbb{1}\{\mathcal{E}\} \sum_{x \in C \setminus C'} R_x\right] = \mathbb{E}\left[\sum_{x \in C \setminus C'} w_C(x) \left(\sum_{i=1}^{n} r_i(x) \|c_i^* - x\|^p + r_i(x) \|\pi(c_i^*) - \pi(x)\|^p\right)\right]$$

$$\leq 2\alpha \operatorname{cost}_p(\mathcal{C}^*) + (1+\varepsilon)^p \cdot 2\alpha \operatorname{cost}_p(\mathcal{C}^*) + \alpha \operatorname{cost}_p(\mathcal{C}^*)$$

$$\leq 5(1+\varepsilon)^p \alpha \operatorname{cost}_p(\mathcal{C}^*).$$

By Markov's inequality, we have that

$$\mathbf{Pr}\left[\varepsilon^{1-p}\mathbb{1}\{\mathcal{E}\}\sum_{x\in C\setminus C'}R_x \geq c_{\eta\varepsilon\alpha}\mathrm{cost}_p(\mathcal{C}^*)\right] \leq \varepsilon^{1-p}\cdot\frac{5(1+\varepsilon)^p\alpha\mathrm{cost}_p(\mathcal{C}^*)}{c_{\eta\varepsilon\alpha}\mathrm{cost}_p(\mathcal{C}^*)} = \eta.$$

Recalling that bounds (13) and (14) hold with probability at least $1-\binom{n}{2}\delta$, by a union bound we have

$$\mathrm{cost}_p(\pi(C)) \leq A(\mathrm{cost}_p(C) + c_{\eta\varepsilon\alpha}\mathrm{cost}_p(\mathcal{C}^*))$$
$$\mathrm{cost}_p(C) \leq A(\mathrm{cost}_p(\pi(C)) + c_{\eta\varepsilon\alpha}\mathrm{cost}_p(\mathcal{C}^*)),$$

with probability at least $1-\eta-\binom{n}{2}\delta$. □

We now show that the cost of any valid solution to the Wasserstein barycenter problem (again in the sense of Definition 2.1) is roughly preserved under a random $(\varepsilon,\delta,\alpha)$-standard dimensionality reduction map. The proof is similar to Theorem 3.5 in [MMR19], but we again use a weighted version of the distortion graphs.

**Theorem B.13.** *Let $\mu_1,\ldots,\mu_k$ be $k$ discrete distributions with support size $n$ on $\mathbb{R}^d$, and let $X = \bigcup_{i=1}^k \mathrm{supp}(\mu_i)$. Given $\varepsilon \in (0,1/4)$ and $\delta \in (0,1)$ and $\pi : \mathbb{R}^d \to \mathbb{R}^m$ with*

$$m = O\left(\frac{\log\frac{n}{\delta} + p\log\frac{1}{\varepsilon} + p^2}{\varepsilon^2}\right),$$

*then with probability at least $1-\delta$, we have that simultaneously for every solution $\mathcal{C} = (C_1,\ldots,C_n)$ and corresponding weight functions $w_j(\cdot)$ of $X$,*

$$\mathrm{cost}_p(\pi(\mathcal{C})) \leq (1+\varepsilon)^{3p}\mathrm{cost}_p(\mathcal{C})$$
$$(1-\varepsilon)\mathrm{cost}_p(\mathcal{C}) \leq (1+\varepsilon)^{3p-1}\mathrm{cost}_p(\pi(\mathcal{C})).$$

*Proof.* We define $\theta = \min(\varepsilon^{p+1}3^{-(p+1)(p+2)},\delta\varepsilon^p/(10n(1+\varepsilon)^{4p-1},1/10^{p+1})$. Let $\delta' \leq \min(\theta^7/600,\theta/n)$, $\binom{n}{2}\delta' \leq \delta/2$, $\alpha \leq \delta$, and $\eta \leq \frac{\delta}{2}$. Let the constant in $m$ be sufficiently large, so that $\pi$ is a random $(\varepsilon,\delta',\alpha)$-standard dimensionality reduction map. Note that the constant $m$ is independent of the quantities $\varepsilon,\delta',\alpha$ due to the properties of a standard dimensionality reduction map. Then we seek to apply Theorem B.12 and note that with these values of $\varepsilon,\delta'$, and $\alpha$, we have $A \leq (1+\varepsilon)^{3p-1}$ and $c_{\eta\varepsilon\alpha} \leq \varepsilon/n$.

Let $\mathcal{E}$ be the event that we have

$$\mathrm{cost}_p(\pi(\mathcal{C})) \leq (1+\varepsilon)^{3p}\mathrm{cost}_p(\mathcal{C}) \tag{17}$$
$$(1-\varepsilon)\mathrm{cost}_p(\mathcal{C}) \leq (1+\varepsilon)^{3p-1}\mathrm{cost}_p(\pi(\mathcal{C})) \tag{18}$$

simultaneously any valid solution $\mathcal{C}$ to the Wasserstein barycenter problem. Suppose that the event $\mathcal{E}$ does not occur, so that there exists a solution $\mathcal{C} = \{C_1,\ldots,C_n\}$ and corresponding weight functions $w_j(\cdot)$ that violates (17) or (18). If (17) fails to hold, then

$$\sum_{i=1}^n \mathrm{cost}_p(\pi(C_i)) = \mathrm{cost}_p(\pi(\mathcal{C}))$$
$$\geq (1+\varepsilon)^{3p}\mathrm{cost}_p(\mathcal{C})$$
$$\geq ((1+\varepsilon)^{3p-1}+\varepsilon)\mathrm{cost}_p(\mathcal{C})$$
$$\geq A\left(\sum_{i=1}^n \mathrm{cost}_p(C_i)\right) + \varepsilon\mathrm{cost}_p(\mathcal{C}).$$

Similarly, if (18) fails to hold, then

$$\sum_{i=1}^n \mathrm{cost}_p(C_i) = (1-\varepsilon)\mathrm{cost}_p(\mathcal{C}) + \varepsilon\mathrm{cost}_p(\mathcal{C})$$
$$\geq (1+\varepsilon)^{3p-1}\mathrm{cost}_p(\pi(\mathcal{C})) + \varepsilon\mathrm{cost}_p(\mathcal{C})$$
$$\geq A\left(\sum_{i=1}^n \mathrm{cost}_p(\pi(C_i))\right) + \varepsilon\mathrm{cost}_p(\mathcal{C}).$$

It follows that there exists some $i \in [n]$ such that at least one of the following inequalities holds:

$$\text{cost}_p(\pi(C_i)) \geq A \, \text{cost}_p(C_i) + \frac{\varepsilon}{n} \, \text{cost}_p(\mathcal{C})$$

$$\text{cost}_p(C_i) \geq A \, \text{cost}_p(\pi(C_i)) + \frac{\varepsilon}{n} \, \text{cost}_p(\mathcal{C}).$$

Let $\mathcal{C}^*$ be the optimal solution (i.e. the actual Wasserstein barycenter) for $\mu_1, \ldots, \mu_k$. In particular, this means that $\text{cost}_p(\mathcal{C}^*) \leq \text{cost}_p(\mathcal{C})$. Then one of the following inequalities must hold:

$$\text{cost}_p(\pi(C_i)) \geq A \, \text{cost}_p(C_i) + \frac{\varepsilon}{n} \, \text{cost}_p(\mathcal{C}^*)$$

$$\text{cost}_p(C_i) \geq A \, \text{cost}_p(\pi(C_i)) + \frac{\varepsilon}{n} \, \text{cost}_p(\mathcal{C}^*).$$

By Theorem B.12, one of these inequalities can hold with probability at most $\eta + \binom{n}{2}\delta'$. Since $\eta \leq \frac{\delta}{2}$ and $\binom{n}{2}\delta' \leq \frac{\delta}{2}$, it follows that $\mathcal{E}$ occurs with probability at least $1 - \delta$. $\qquad\square$

Finally, we show how to rescale the parameters of Theorem B.13 to prove Theorem 4.1.

*Proof of Theorem 4.1.* Observe that Theorem B.13 with a rescaling of $\varepsilon' := (1 + \varepsilon)^{1/(3p)-1} = O(\varepsilon/p)$ immediately implies the desired claim. $\qquad\square$

**Fast dimensionality reduction.** The dimensionality reduction maps of Theorem B.13 generally require multiplication by a dense matrix of (scaled) subgaussian random variables. Thus for $\delta = O(1)$ and $p = O(1)$, applying the dimensionality reduction map using rectangular matrix multiplication takes $O\left(\frac{dkn \log n}{\varepsilon^2}\right)$ time. We provide a tradeoff between runtime and dimension using the following observation:

**Theorem B.14.** *[MMR19] There exists a family of $(\varepsilon, \delta, \alpha)$-dimensionality reduction maps $\pi : \mathbb{R}^d \to \mathbb{R}^m$ with $m = O\left(\frac{p^6}{\varepsilon^2} \log^2 \frac{k}{\varepsilon}\delta\right)$ with runtime $O(d \log d)$ on an input vector $v \in \mathbb{R}^d$.*

We describe the construction of $\pi$ in Theorem B.14 as in [AC09, MMR19]. By a standard padding with zeros argument, we first assume that $d$ is a power of two. We define $D$ to be a diagonal $d \times d$ matrix with i.i.d. uniform signs, i.e., $\pm 1$ entries. We define $H$ to be a normalized Hadamard transform so that $H$ is an orthogonal matrix with all entries $\pm\frac{1}{\sqrt{d}}$ and $Hv$ can be computed in $O(d \log d)$ time.

We define $S$ to be a diagonal "sampling" matrix with i.i.d. entries, so that $\mathbf{Pr}\left[S_{i,i} = \frac{\sqrt{d}}{m}\right] = \frac{m}{d}$ and $\mathbf{Pr}\left[S_{i,i} = 0\right] = 1 - \frac{m}{d}$. Let $\Pi : \mathbb{R}^d \to \mathbb{R}^d$ be defined by $\Pi = SHD$ and note that the expected dimension of the image of $\Pi$ is $m$. Then $\pi$ is defined to be the image of $\Pi$, conditioned on the event that the dimension of the image of $\Pi$ is at most $10d$ [AC09, MMR19]. Hence, we obtain the following fast dimensionality reduction:

**Corollary B.15** (Fast dimensionality reduction). *Let $\mu_1, \ldots, \mu_k$ describe an instance of the Wasserstein barycenter problem with the $L_p$ objective in $\mathbb{R}^d$. Given $\varepsilon \in (0, 1/4)$ and $\delta \in (0, 1)$, there exists $\pi : \mathbb{R}^d \to \mathbb{R}^m$ with*

$$m = O\left(\frac{p^6}{\varepsilon^2} \log^2 \frac{k}{\varepsilon}\delta\right),$$

*that uses $O(d \log d)$ runtime to apply the mapping to each point and provides the same guarantees as Theorem B.13.*

**Alternative proof for Theorem 4.1.** An anonymous NeurIPS 2021 reviewer pointed out the following alternative proof for Theorem 4.1. Given the $k$ distributions $\mu_1, \ldots, \mu_k$ in $\mathbb{R}^d$, consider each distribution $\mu_i$ as a multiset $U_i$ of $\mathbb{R}^d$ of size $M$ for a sufficiently large $M$. Then the Wasserstein barycenter problem can be rewritten as the optimization problem

$$\min \sum_{i=1}^{k} \frac{1}{M} \sum_{u \in U_i} \|u - c_{f_i(u)}\|_2,$$

for an assignment function $f_i : U_i \to [n]$, subject to the constraints:

1. $c_1, \ldots, c_n \in \mathbb{R}^d$
2. $\frac{1}{M} \sum_{u \in U_i} \mathbf{1}[f_i(u) = j] = a_j$, where $\sum_{j \in [n]} a_j = 1$.

Setting $U$ to be the multi-set defined by the union of all $U_i$ with $i \in [k]$ and $f : U \to [n]$ defined by $f(u) = f_i(u)$ for $u \in U_i$, the above optimization can be further rewritten as

$$\min \frac{1}{M} \sum_{u \in U} \|u - c_{f(u)}\|_2,$$

subject to the same constraints. Since this is a constrained $k$-median clustering problem and [MMR19] show that the cost of *every* clustering is preserved to within a $(1 + \varepsilon)$-factor under a projection to $O\left(\frac{1}{\varepsilon^2} \log n\right)$ dimensions, then the optimal clustering under the above constraints are also preserved to within a $(1 + \varepsilon)$-factor.

## C   Proofs for Section 5

We need the following theorem which relates the size of coresets, obtained from importance sampling according to sensitivity values, to the pseudo-dimension of a related function class. Sensitivity sampling has been used to design coresets for many problems in machine learning such as support vector machine, Gaussian mixture models, projective clustering, principal component analysis, $M$-estimators, Bayesian logistic regression, and generative adversarial networks, e.g., see recent surveys on coresets such as [BLK17, Fel20].

From Theorem 5.5, we now need to bound the following two things to obtain a coreset.

1. The total sensitivity $S$,
2. The pseudo-dimension of $\mathcal{F}$.

(Note that Theorem 5.2 in [BLK17, Lan18] is stated for coresets of general functions, we specialize it to the case of Wasserstein Barycenters.)

We begin by bounding (1), the total sensitivity $S$. To do so, we need to define a function $s$ that informs how we sample the distributions in $M$. The following lemma shows that it suffices to use a constant factor approximation to the best barycenter solution to perform the sampling.

**Lemma C.1.** *Let $\alpha, p \geq 1$ and let $\nu'$ be an $\alpha$-approximate solution to the $p$-Wasserstein Barycenter problem for the set $M$ of distributions in $\mathbb{R}^d$ with support size at most $n$. That is,*

$$\frac{1}{|M|} \sum_{\mu \in M} W(\mu, \nu')^p \leq \alpha \frac{1}{|M|} \sum_{\mu \in M} W(\mu, \nu^*)^p$$

*where $\nu^*$ is the optimal barycenter distribution. Then the sensitivity $\sigma(\mu)$ for $\mu \in M$ defined as in Definition 5.3 is bounded by*

$$\sigma(\mu) \leq s(\mu) \leq \frac{\alpha 2^{p-1} W(\mu, \nu')^p}{\frac{1}{|M|} \sum_{\widetilde{\mu} \in M} W(\widetilde{\mu}, \nu')^p} + \alpha 4^{p-1} + 4^{p-1}.$$

*Furthermore, it holds that*

$$\mathfrak{S} \leq \alpha(4^{p-1} + 2^{p-1}) + 4^{p-1}.$$

**Remark C.2.** *Note that $p$ is typically $O(1)$; for example $p = 1$ or $p = 2$ are the most common choices.*

Now our goal is to bound (2), the pseudo-dimension of the function class $\mathcal{F}$ in Theorem 5.5. First we relate pseudo-dimension to VC dimension of related threshold functions and then we state a result relating the VC dimension to the algorithmic complexity of computing these threshold functions.

**Lemma C.3** (Pseudo-dimension to VC dimension, Lemma 10 in [LFKF18]). *For any $f \in \mathcal{F}$, let $B_f$ be the indicator function of the region on or below the graph of $f$, i.e., $B_f(x, y) = sgn(f(x) - y)$. The pseudo-dimension of $\mathcal{F}$ is equivalent to the VC-dimension of the subgraph class $B_{\mathcal{F}} = \{B_f \mid f \in \mathcal{F}\}$.*

Then we need the following theorem that relates VC dimension of a function class to its computational complexity.

**Lemma C.4** (Theorem 8.14 in [AB99]). *Let $h : \mathbb{R}^a \times \mathbb{R}^b \to \{0, 1\}$, determining the class*

$$\mathcal{H} = \{x \to h(\theta, x) : \theta \in \mathbb{R}^a\}.$$

*Suppose that any $h$ can be computed by an algorithm that takes as input the pair $(\theta, x) \in \mathbb{R}^a \times \mathbb{R}^b$ and returns $h(\theta, x)$ after no more than $t$ of the following operations:*

- *arithmetic operations $+, -, \times$, and $/$ on real numbers,*

- *jumps conditioned on $>, \geq, <, \leq, =$, and $=$ comparisons of real numbers, and*

- *output $0, 1$,*

*then the VC dimension of $\mathcal{H}$ is $O(a^2 t^2 + t^2 a \log a)$.*

Combining the previous lemmas lets us prove the following theorem. At a high level, we are instantiating Lemma C.4 with the complexity of *computing* any function in the function class $\mathcal{F}$ defined in 5.5. This is a similar proof strategy used in [LFKF18] to control the coreset size of a different problem (coresets for Gaussian mixture models) in the proof of their Theorem 2. However, there is a flaw in their argument as we believe that they incorrectly apply the pseudo-dimension argument to a slightly different function class. We propose a fix in our proof below.

Finally note that the function class $\mathcal{F}$ depends on the the sampling function $s$. For our purposes, we use the sampling function defined in Lemma C.1 which samples according to a fixed $O(1)$ approximate solution.

**Theorem C.5.** *Consider the set of functions $\mathcal{F}$ defined in Theorem 5.5 for $s : M \to \mathbb{R}$ as defined in Lemma C.1. The pseudo-dimension of $\mathcal{F}$ is $O(n^8 d^2)$.*

Altogether, we can prove the following bound on the size of coresets for the Wasserstein Barycenter problem which states that $k$, the number of distributions, can be reduced to $\mathrm{poly}(n, d)$ for constant $p$.

**Theorem C.6.** *Let $\delta, \varepsilon \in (0, 1)$. Let $\nu'$ be an $\alpha$ approximation to the $p$-Wasserstein Barycenter problem for a set $M$ of distributions in $\mathbb{R}^d$ with support size at most $n$ and let $s : M \to \mathbb{R}$ be defined as in Lemma C.1. Consider sampling a subset $K \subseteq M$ of size $\widetilde{\Omega}(\alpha 4^{p-1} n^8 d^4 / \varepsilon^2)$ where $\widetilde{\Omega}$ hides logarithmic factors. Then $K$ satisfies Definition 5.1 with probability $1 - \delta$.*

*Proof.* The result follows from instantiating Theorem 5.5 with the bound of $S$ from C.1 and the pseudo-dimension bound in C.5. $\square$

**Remark C.7.** *Again we remark that we are not optimizing for the exact constants in the exponents in Theorem C.6. There are several places where such optimizations can possibly be made. For example, using a faster algorithm than the Hungarian algorithm to argue about the pseudo-dimension bound in Theorem C.5. However, any such optimizations would result in coresets of size $\mathrm{poly}(n, d)$ if we are to use the sensitivity sampling framework.*

*Proof of Lemma C.1.* Let $\nu$ denote an arbitrary barycenter distribution. For any $\mu \in M$, the triangle inequality gives us

$$W(\mu, \nu)^p \leq 2^{p-1}(W(\mu, \nu')^p + W(\nu', \nu)^p)$$

where we have used the fact that $(x + y)^p \leq 2^{p-1}(x^p + y^p)$ for non-negative $x, y$ and $p \geq 1$. Using a similar reasoning, we have

$$W(\nu', \nu)^p \leq 2^{p-1}(W(\nu', \mu') + W(\mu', \nu))$$

for every $\mu' \in M$. Averaging over all $\mu'$ gives us

$$W(\nu', \nu)^p \leq \frac{2^{p-1}}{|M|} \sum_{\mu' \in M} (W(\nu', \mu')^p + W(\mu', \nu)^p).$$

It follows that

$$
\frac{W(\mu,\nu)^p}{\frac{1}{|M|}\sum_{\widetilde{\mu}\in M}W(\widetilde{\mu},\nu)^p} \leq \frac{2^{p-1}W(\mu,\nu')^p}{\frac{1}{|M|}\sum_{\widetilde{\mu}\in M}W(\widetilde{\mu},\nu)^p} + \frac{\frac{4^{p-1}}{|M|}\sum_{\mu'\in M}\left(W(\nu',\mu')^p + W(\mu',\nu)^p\right)}{\frac{1}{|M|}\sum_{\widetilde{\mu}\in M}W(\widetilde{\mu},\nu)^p}
$$

$$
\leq \frac{\alpha 2^{p-1}W(\mu,\nu')^p}{\frac{1}{|M|}\sum_{\widetilde{\mu}\in M}W(\widetilde{\mu},\nu')^p} + \frac{\frac{\alpha 4^{p-1}}{|M|}\sum_{\mu'\in M}W(\nu',\mu')^p}{\frac{1}{|M|}\sum_{\widetilde{\mu}\in M}W(\widetilde{\mu},\nu')^p} + 4^{p-1}
$$

$$
\leq \frac{\alpha 2^{p-1}W(\mu,\nu')^p}{\frac{1}{|M|}\sum_{\widetilde{\mu}\in M}W(\widetilde{\mu},\nu')^p} + \alpha 4^{p-1} + 4^{p-1} =: s(\mu)
$$

where we have used the fact that $\nu'$ is an $\alpha$-approximation to the optimal barycenter and thus,

$$
\sum_{\widetilde{\mu}\in M}W(\widetilde{\mu},\nu)^p \geq \sum_{\widetilde{\mu}\in M}W(\widetilde{\mu},\nu^*)^p \geq \frac{1}{\alpha}\sum_{\widetilde{\mu}\in M}W(\widetilde{\mu},\nu')^p
$$

by assumption on $\nu^*$ and $\nu'$. This gives us

$$
\mathfrak{S} \leq \frac{1}{|M|}\sum_{\mu\in M}s(\mu) \leq 4^{p-1} + \alpha 4^{p-1} + \frac{\alpha 2^{p-1}}{|M|}\frac{\sum_{\mu\in M}W(\mu,\nu')^p}{\frac{1}{|M|}\sum_{\widetilde{\mu}\in M}W(\widetilde{\mu},\nu')^p} = \alpha(4^{p-1}+2^{p-1})+4^{p-1}.
$$

Since $\mu$ and $\nu$ were arbitrary, the result follows. $\qquad\square$

*Proof of Theorem C.5.* Let $\nu \in N$ where $N$ is the set of all possible barycenter distributions with support size $n$, as defined in Theorem 5.5 and let $r \in \mathbb{R}$. Let $M$ be a set of $k$ different distributions on $\mathbb{R}^d$, each with support size at most $n$. Then for $x \in M$, we define $h : N \times \mathbb{R} \times M \to \{0,1\}$ by $h(\nu,r,x) = h_{\nu,r}(x) = \mathbb{1}\{W(x,\nu)^p/s(x) \geq r\}$.

We remark that the conceptually similar proof of Theorem 2 in [LFKF18] used to bound the coreset sizes of Gaussian mixture models erroneously omits the function $s(x)$ in the definition of $h$ above.

Now let the corresponding function class $\mathcal{H}$ be defined as

$$
\mathcal{H} = \{h_{\nu,r} : M \to \{0,1\} \mid \nu \in N, r \in \mathbb{R}\}.
$$

Note that computing $W(x,\nu)^p$ is equivalent to computing the minimum cost bipartite matching between the weighted points of $x$ and $\nu$ with edge costs coming from the Euclidean metric raised to the $p$th power. By the well known Hungarian algorithm, this can be computed in $O(n^3 + n^2d)$ arithmetic steps where the first term is from the Hungarian algorithm and the second term is to compute the edge costs between $x$ and $\nu$. Furthermore, computing $s(x)$ can also be done in $O(n^3 + n^2d)$ since we need to find the cost of the matching between $x$ and $\nu'$ where $\nu'$ is the approximate solution used to define $s$ in Lemma C.1 (the other terms of $s(x)$ are constant).

Therefore by Lemma C.4, the VC dimension of $\mathcal{H}$ is at most $O((nd)^2 \cdot (n^3 + n^2d)^2) = O(n^8d^4)$ since we need $O(nd)$ variables to define $x$ and $\nu$. Now note that the function class $\mathcal{H}$ is equivalent to the function class $\mathcal{J}$ defined as

$$
\mathcal{J} = \{f_{\nu,r} : M \to \{0,1\} \mid \nu \in N, r \in \mathbb{R}\}
$$

where

$$
f_{\nu,r} = \mathbb{1}\left\{\frac{W(\cdot,\nu)^p}{s(\cdot)\sum_{m\in M}W(m,\nu)^p} \geq r\right\}
$$

This is because we are letting $r$ range over all all the reals in the definition of $\mathcal{H}$. Therefore it also follows that the VC dimension of class $\mathcal{J}$ is $O(n^8d^4)$. Finally by Lemma C.3, the pseudo-dimension of $\mathcal{F}$ as defined in Theorem 5.5 can be bounded by $O(n^8d^4)$. $\qquad\square$

# D   Lower Bound Proofs for Section 4.1

We now turn to proving lower bounds, showing that our dimensionality reduction is optimal up to constant factors. To begin, we first need two auxiliary results regarding random linear transformations.

The proof of Theorem 9 in [KMN11] states the following:

**Theorem D.1.** *[KMN11] Let $M : \mathbb{R}^d \to \mathbb{R}^m$ be a linear transformation with $d > 2m$ and $\varepsilon > 0$ sufficiently small. Then for a randomly chosen unit vector $u \in \mathbb{R}^d$*

$$\mathbf{Pr}\left[\|Mu\|^2 < 1 - \varepsilon\right] \geq \exp(-O(m\varepsilon^2 + 1))).$$

By applying rotational invariance of a standard Gaussian, we arrive at the following corollary of Theorem D.1.

**Corollary D.2.** *Let $M : \mathbb{R}^d \to \mathbb{R}^m$ be a random matrix with i.i.d. entries from $\mathcal{N}(0, \sigma)$, $d > 2m$ and $\varepsilon > 0$ sufficiently small. Then for any vector $u \in \mathbb{R}^d$,*

$$\mathbf{Pr}\left[\|Mu\|^2 < 1 - \varepsilon\right] \geq \exp(-O(m\varepsilon^2 + 1))).$$

We are now equipped to prove Theorem 4.2, restated here for convenience.

**Theorem 4.2.** *Consider the setup of Theorem 4.1. Any Gaussian matrix used as a dimension reduction map that allows a $(1 + \varepsilon)$-approximation to the optimal Wasserstein barycenter requires dimension $\Omega(\log n/\varepsilon^2)$.*

*Proof.* Let $t > 0$ be a parameter and $N > 0$ be a sufficiently large constant. Consider the points $S = p_1, p_2, \ldots, p_t, q_1, q_2, \ldots, q_t$ so that $p_i = Ne_i$ and $q_i = (N+1)e_i$ for each $i \in [t-1]$, where $e_i$ is the $i$-th elementary vector. Let $p_t = Ne_t$ and $q_t = (N + 1 - C\varepsilon)e_t$ for a parameter $C > 0$. Hence we have:

1. $\|p_i - q_i\| = 1$ for each $i \in [t-1]$.

2. $\|p_t - q_t\| = 1 - C\varepsilon$.

3. $\min_{i \neq j}(\|p_i - p_j\|, \|p_i - q_j\|, \|q_i - q_j\|) \geq N(1 - C\varepsilon)\sqrt{2}$.

Let $n = 2t - 1$ and consider the $k = 2t$ distributions $\mu_1, \ldots, \mu_{2t}$ so that for each $i \in [t]$, $\mu_i$ has weight $\frac{1}{2t-1}$ on each of the points in $S$ except $p_i$, at which it has weight zero. Similarly, for each $i \in [t]$, suppose $\mu_{i+t}$ has weight $\frac{1}{2t-1}$ on each of the points in $S$ except $q_i$, at which it has weight zero. Thus, the total weight across all distributions at each of the $2t$ points in $S$ is exactly 1. It can easily be shown that the barycenter of support size at most $n$ has cost $(1 - C\varepsilon)^p$, by choosing the points $p_i$ and $q_i$ for each $i \in [t-1]$ and then either $p_t$ or $q_t$.

We now show that for a Gaussian matrix $M$ with dimension $m = \left(\frac{\log n}{1000\varepsilon^2}\right)$, with high probability there exists some $j \in [t-1]$ such that $\|Mp_j - Mq_j\| \leq \|Mp_t - Mq_t\|$. First note that $\|Mp_t - Mq_t\|$ equals $(1 - C\varepsilon)$ times a random variable that follows a Chi-squared distribution with $m$ degrees of freedom. By standard concentration inequalities on the sum of $m$ independent $\chi^2$ variables, e.g., Equation 2.21 in [Wai19], we have that

$$\mathbf{Pr}\left[\chi_m^2 \leq (1 - \varepsilon)m\right] \leq \exp\left(-\frac{m\varepsilon^2}{8}\right) \leq 0.01.$$

Thus the probability that $\|Mp_t - Mq_t\| \leq (1 - C\varepsilon)(1 - \varepsilon)$ is at most 0.01.

Moreover we have that for $i \neq [t]$, $p_i - q_i$ is a unit vector in $\mathbb{R}^d$. Thus by Corollary D.2, we have that

$$\mathbf{Pr}\left[\|Mp_i - Mq_i\|^2 < 1 - (4C)\varepsilon\right] \geq \exp(-O(m(4C)\varepsilon^2 + 1)) \geq \frac{1}{n^{1/5}},$$

for $m = \left(\frac{\log n}{1000C^2\varepsilon^2}\right)$. Therefore since $t = \Omega(n)$, we have that there exists $i \neq [t-1]$ with $\|Mp_i - Mq_i\|^2 < 1 - (4C)\varepsilon$ with probability at least 0.99.

Hence with probability at least 0.98, the optimal clustering in the projected space will be the projection of the points $p_1, \ldots, p_{j-1}, p_{j+1}, \ldots, p_t$, the points $q_1, \ldots, q_{j-1}, q_{j+1}, \ldots, q_t$, and either the point $p_j$ or $q_j$. Thus in this case, the corresponding cost in the original space is exactly 1, so that the dimension reduction map does not allow a $(1 + \varepsilon)$-approximation to the optimal clustering. $\qquad\square$

Next, we turn to proving lower bounds on dimensionality reduction for the optimal transport problem. Again, we first need an auxiliary concentration result for high-dimensional Gaussians.

**Lemma D.3.** *Let $C \geq 1$ and fix some point $v$ of norm at most $C$ in $\mathbb{R}^d$. Then, if $x \sim \frac{1}{\sqrt{d}} \cdot \mathcal{N}(0, I_d)$ is a $d$-dimensional scaled multivariate Normal, then $\Pr(\|x - v\| \leq \frac{1}{C}) \geq n^{-1/10}$, if $d \leq \log n/(10C^2)$ and $n$ is sufficiently large.*

*Proof of Lemma D.3.* By the rotational symmetry of the multivariate normal, assume $v = (r, 0, \ldots, 0)$, where $0 \leq r \leq C$. Then, if $x = (x_1, y)$ for $x_1 \in \mathbb{R}, y \in \mathbb{R}^{d-1}$, then if $r - \frac{1}{2C} \leq x_1 \leq r$ and $\|y\| \leq \frac{1}{2C}$, then we indeed have $\|x - v\| \leq \frac{1}{C}$. Since $\sqrt{d} x_1 \sim \mathcal{N}(0,1)$ and $r \leq C$, the probability that $r - \frac{1}{2C} \leq x_1 \leq r$ equals the probability that $\mathcal{N}(0,1) \in [(r - 1/2C)\sqrt{d}, r\sqrt{d}]$, which is at least $\frac{\sqrt{d}}{2C} \cdot \frac{1}{\sqrt{2\pi}} \cdot e^{-C^2 d/2}$. Moreover, the probability that $\|y\| \leq \frac{1}{2C}$ is at least $\left(\frac{1}{2eC}\right)^d$. Therefore,

$$\Pr\left(\|x - v\| \leq \frac{1}{C}\right) \geq \frac{\sqrt{d}}{2C} \cdot \frac{1}{\sqrt{2\pi}} \cdot e^{-C^2 d/2} \cdot \left(\frac{1}{2eC}\right)^d$$
$$\geq n^{-1/10},$$

where the last inequality is true because $d \leq \log n/(10C^2)$ and that $n$ is sufficiently large. $\square$

*Proof of Theorem 4.4.* Our point sets will be $A \cup B = \{e_i\} \cup \{e_i/2\}$ with the property that $e_i$ and $e_i/2$ will be in different sets and we will alternate the $i$'s such that $e_i \in A$. The optimal matching in $\mathbb{R}^d$ is to match each $e_i$ to $e_i/2$ leading to cost $d/2$.

Our strategy is to show that if we project $A \cup B$ to $m = o(\log d)$ dimensions, then we can find a matching of cost $o(d)$. Towards that end, let $C = \sqrt{\log d/10m} = \omega(1)$ and let $\pi$ be a random projection to $o(\log d)$ dimensions. First, we will show that points $\pi e_i$ with $\|\pi e_i\| \leq C$ will have 'many' other points $\pi e_j$ sufficiently near by so that we can match $\pi e_i$ to $\pi e_j$ (assuming they are in different sets). We then show that the points with $\|\pi e_i\| \geq C$ can be disregarded.

More formally, by Lemma D.3, the number of other points $e_j$ such that $\|\pi e_i - \pi e_j\| \leq 1/C$ and $e_j$ is in a different set than $e_i$ is a binomial random variable $B(d - 1, q)$ where $q \geq d^{-1/10}/2$. Therefore the number of such $j$'s is at least $d^{c'}$ for some constant $c' > 0$ except with probability at most $\ll 1/d$. By a union bound, we can assume that every $i$ such that $\|\pi e_i\| \leq C$ has at least $d^{c'}$ other $\pi e_j$'s such that $\|\pi e_i - \pi e_j\| \leq 1/C$ and $e_j$ is in a different set than $e_i$. Now consider the following greedy matching procedure to match the points $e_i$ with $\|\pi e_j \leq C\|$ which may not be optimal: for every such $\pi e_i$, we try to match it to any $\pi e_j$ that is within distance $1/C$ greedily (we also map $\pi e_i/2$ to $\pi e_j/2$). We do this until it is no longer possible. Then, we try to match each $\pi e_i$ to some $\pi e_j$ within distance $2/C$ greedily until no longer possible. Then, we just match $\pi e_i$ to $\pi e_i/2$. Note that every possible match contributes $O(1/C)$ to the matching cost so altogether, this greedy matching has cost at most $O(d/C) = o(d)$.

We now want to show that not many of the $\pi e_i$ will be leftover that have to be matched to $\pi e_i/2$. Consider maximally covering the set of all such $\pi e_i$ that have to be matched to $\pi e_i/2$ with disjoint balls of radius $1/C$. First, every such $\pi e_i$ must be in some ball since other wise, it would have been within radius $2/C$ of some $\pi e_{i'}$ and we would have matched them. Now each ball intersects with at least $d^{c'}$ other points in $A \cup B$ by our calculation in the previous paragraph. Therefore, there can be at most $O(d^{1-c'})$ such balls and hence, the matching cost induced by these points is at most $O(Cd^{1-c'}) = o(d)$ as well.

Now we just have to deal with points $e_i$ that satisfy $\|\pi e_i\| \geq C$. If they are not matched already, we just match them to $\pi e_i/2$. The expected cost incurred by one of these edges in the matching is

$$\mathbb{E}\left[\|Ge_i\| \cdot \mathbf{1}_{\|Ge_i\| \geq C}\right] \leq \sqrt{\mathbb{E}\left[\|Ge_i\|^2\right] \cdot \Pr(\|Ge_i\| \geq C)}$$
$$\leq \sqrt{1 \cdot \exp\left(-m \cdot (C-1)^2/8\right)}$$
$$\leq \exp\left(-(C-1)^2/16\right) \leq \frac{1}{C},$$

so the total expected cost from these edges is at most $O(d/C) = o(d)$. Finally by an application of Markov's inequality and a union bound, we have that with probability at least $2/3$, we can find a matching in $\mathbb{R}^m$ with cost at most $o(d)$ and hence, the optimal matching in the projected space has cost at most $o(d)$, as desired. $\square$

If $x \in S^{m-1}$ and $\pi$ is an appropriately normalized Gaussian dimensionality reduction map, then the following statements hold about the distribution of $\|\pi x\|$ [IN07]:

$$\Pr(|\|\pi x\| - 1| \geq t) \leq \exp(-dt^2/8), \tag{19}$$

$$\Pr(\|\pi x\| \leq 1/t) \leq \left(\frac{3}{t}\right)^d. \tag{20}$$

Finally, we prove Theorem 4.3. Note that Theorem 4.4 states that after we perform a random projection to $o(\log n)$ dimensions, the *cost* (i.e., the actual objective numerical value) of the optimal matching in the projected space will be much smaller than the cost of the optimal matching in the original dimension. This highlights that if we just wish to approximate the cost of the matching, we cannot do better than the standard JL lemma dimension bound. Note that given Theorem 4.3 it is still possible that the optimal matching in the projected dimension is approximately equal to the optimal matching in the original dimension since Theorem 4.3 is only addressing the cost. We show in the proof of Theorem 4.3 that this is not the case; the optimal matching in the projected space will induce a poor matching in the original dimension if we project to much fewer than $\log n$ dimensions.

*Proof of Theorem 4.3.* Many details of this proof follow similarly as in the proof of Theorem 4.4. Let $C = \sqrt{\log d / 10m} = \omega(1)$. Our point sets will be $A \cup B = \{e_i \cdot k/C\}$ for all $1 \leq i \leq d$ and $1 \leq k \leq C$. We refer to $A$ and $B$ as "classes" and assume that $C$ is an even integer. The partition of the points is as follows. For a fixed $i$, the points $e_i \cdot k/C$ will alternate which set they belong to, i.e, $e_i/C$ will be in $A$, $2e_i/C$ will be in $B$ etc. We will also impose the condition that half of the $e_i$'s will be in $A$ and the other half will be in $B$. Now note that the optimal matching in $\mathbb{R}^d$ is to just match each $e_i \cdot k/C$ to $e_i \cdot (k+1)/C$ for $1 \leq k \leq C-1$ which results in matching cost $O(d)$.

Now consider a random projection $\pi$ to $m = o(\log d)$ dimensions. Our strategy is to show that the optimal matching in $\mathbb{R}^m$ will contain many edges between different $e_i$'s which will induce a large matching cost in $\mathbb{R}^d$.

Towards that end, define 'level $k$' to be the set of points of the form $e_i \cdot k/C$ for some $i$. First note that if $\pi$ is a Gaussian random projection, we have that $\|\pi e_i\| \in [1/10, 100]$ with probability at least $1 - \exp(-m/10) - (3/100)^m > 0.06$ from equations (19) and (20). Thus we can say by a standard Chernoff bound that a $\Theta(1)$ fraction of $e_i$ will satisfy $\|\pi e_i\| = \Theta(1)$ with exponentially small failure probability. By Lemma D.3, for each such $\pi e_i$, there exists some $e_j$ such that $\|\pi e_i - \pi e_j\| \leq 1/(100C)$ (again up to some exponentially small failure probability). Since the basis vectors are equally partitioned into the two classes, we can further assume that $e_j$ is in a different class than $e_i$.

Let $I$ be the set of $i$'s such that $\|\pi e_i\| = \Theta(1)$ and there is some $j$ such that $\|\pi e_i - \pi e_j\| \leq 1/(100C)$ and $j$ is in a different class. For each $i \in I$, the distance between $\pi e_i \cdot k/C$ and $\pi e_i \cdot \ell/C$ for any $\ell \neq k$ is at least $\frac{1}{10C}$ but the distance between $\pi e_i \cdot k/C$ and $\pi e_j \cdot k/C$ is at most $\frac{1}{100C}$. Thus at all levels, we can potentially switch the matching between $\pi e_i \cdot k/C$ and $\pi e_i \cdot (k+1)/C$ or $\pi e_i \cdot (k-1)/C$ (if it exists) to $\pi e_i \cdot k/C$ and $\pi e_j \cdot k/C$ and the same for the point that $\pi e_i \cdot k/C$ was matched to. Therefore, almost all except possibly 1 of the indices in $I$ across all levels will be matched to a point that comes from a different basis vector. Thus the pullback cost is at least some absolute constant times

$$\sum_{i \in I} \sum_{k=1}^{C} \frac{k}{C} \geq \frac{C}{2} \cdot |I|,$$

which is least $\Omega(C \cdot d) = \Omega(C \cdot M) = \omega(M)$, as desired. $\qquad\square$

# E  Connections to Constrained Low-Rank Approximation

[CEM+15] previously showed that the problem of $k$-means clustering can be formulated as a problem of constrained low-rank approximation, a class of problems which also includes the singular value decomposition (SVD). In this section, we show that the problem of computing a Wasserstein barycenter can be also formulated as a problem of constrained low-rank approximation. Thus efficient subroutines that improve the performance of low-rank approximation can also be used to improve the performance of computing a Wasserstein barycenter.

Recall that for an input matrix $\mathbf{A} \in \mathbb{R}^{a \times b}$ and any set $S$ of rank $c$ orthogonal projection matrices in $\mathbb{R}^{a \times a}$, the goal of constrained low-rank approximation is to find

$$\mathbf{P}^* = \underset{\mathbf{P} \in S}{\operatorname{argmin}} \|\mathbf{A} - \mathbf{P}\mathbf{A}\|_F^2.$$

*Proof of Theorem 6.1.* For each point $x$, let $w_i(x)$ be the weight of $x$ in distribution $w_i$ and for each $j \in [n]$, let $w_{i,j}(x)$ be the weight of $x$ in distribution $w_i$ that is assigned to barycenter $j$, so that we have $\sum_{j \in [n]} w_{i,j}(x) = w_i(x)$ and $\sum_x w_i(x) = 1$ for all $i$. Thus we have the Wasserstein barycenter objective as minimizing

$$\sum_{j \in [n]} \sum_{i \in [k]} w_{i,j}(x) \|x - C_j\|^p.$$

Rewriting the points of $\mu_i$ as $x_{i,1}, x_{i,2}, \ldots, x_{i,n}$, then the Wasserstein barycenter objective for $p = 2$ is

$$\min \sum_{j \in [n]} \sum_{i \in [k]} w_{i,j}(x_{i,j}) \|x_{i,j} - C_j\|^2.$$

Thus we can refold the points $x_{i,j}$ into a matrix of size $\mathbf{A} \in \mathbb{R}^{nk \times d}$ so that the first row of $\mathbf{A}$ consists of the $d$ coordinates of $x_{1,1}$ and more generally row $(i-1)n + j$ of $\mathbf{A}$ consists of the $d$ coordinates of $x_{i,j}$.

Suppose without loss of generality that there exists an integer $N$ such that $w_{i,j}$ is a multiple of $1/N$ for each $i \in [k], j \in [n]$. Let $\mathbf{B} \in \mathbb{R}^{Nk \times d}$ so that each row $(i-1)n + j$ of $\mathbf{A}$ consecutively appears $w_{i,j}(x_{i,j})$ times in $\mathbf{B}$. Thus $\mathbf{B}$ is essentially the matrix whose rows encode each point of each distribution, effectively duplicating each point a number of times equal to its weight in the distribution.

We define a clustering $C = \{C_1, \ldots, C_n\}$ so that there exist weights $w_1, \ldots, w_n$ with $\sum_{j \in [n]} w_j = 1$ with the property that for each $i \in [k]$ and $j \in [n]$, there are exactly $w_j N$ points between rows $(i-1)N + 1$ and $iN$ inclusive are assigned to cluster $j$. Intuitively, this corresponds to each barycenter being assigned weight $w_j$ from each distribution. For each $j \in [n]$, let $\sigma_j$ be the centroid of all the $w_j Nk$ points assigned to $C_j$ and for each $r \in [Nk]$, let $C(r) \in [n]$ be the cluster to which row $r$ is assigned.

Given a clustering $C = \{C_1, \ldots, C_n\}$, we define the cluster indicator matrix $\mathbf{X}_C \in \mathbb{R}^{Nk \times n}$ to be matrix such that row $(i-1)n + j$ in $\mathbf{X}_C$ has entry $\frac{1}{\sqrt{|C_\ell|}}$ in column $\ell \in [n]$ if and only if the corresponding $\frac{1}{N}$ weight of $x_{i,j}$ is assigned to cluster $C_\ell$ (and entry zero otherwise). Thus there exist weights $w_1, \ldots, w_n$ with $\sum_{j \in [n]} w_j = 1$ such that for each $i \in [k]$ and $j \in [n]$, column $j$ has exactly $w_j N$ nonzero entries between rows $(i-1)N + 1$ and $iN$ inclusive. Note in this interpretation, we further have $|C_\ell| = w_j N$.

Since the columns of $\mathbf{X}_C$ have disjoint support, then the corresponding vectors are orthonormal. Thus $\mathbf{X}_C \mathbf{X}_C^\top$ is a rank $n$ projection matrix and we can write the problem of Wasserstein barycenter as the constrained low-rank approximation

$$\min_C \frac{1}{N} \|\mathbf{A} - \mathbf{X}_C \mathbf{X}_C^\top \mathbf{A}\|_F^2 = \sum_{r \in [Nk]} \|\mathbf{B}_r - \sigma_{C(r)}\|^2.$$

Note that the cluster indicator matrix $\mathbf{X}_C$ is constrained to the set of valid clusters $C$ consistent with assignments of the support points in the Wasserstein barycenter to each distribution. $\qquad \square$

# F    NP Hardness of Approximation of Wasserstein Barycenters

In this section, we show the NP-hardness of finding a Wasserstein barycenter with cost within a multiplicative 1.0013 factor of the cost induced by an optimal Wasserstein barycenter. We first the following statement about the hardness of approximation for $k$-means clustering:

**Theorem F.1.** *[LSW17] It is NP-hard to approximate $k$-means clustering within a multiplicative factor of* 1.0013.

The proof of Theorem F.1 relies on a reduction from the Vertex Cover problem on 4-regular graphs. Namely, [CC06] showed that it is NP-hard to distinguish whether a 4-regular graph $G$ with $n$ vertices has vertex cover size at least $A_{\max}n$ or vertex cover at most $A_{\min}n$, for some absolute constants $A_{\min} < A_{\max}$. [LSW17] transformed a 4-regular graph $G$ into a graph $G'$ and embedded $G'$ into $\mathbb{R}^{3n}$ so that for the optimal $k$-means clustering cost of $G'$ (where $k$ is a function of $n$) is at least $C_{\max}$ if the smallest vertex cover of $G$ has size at least $A_{\max}$ and at least $C_{\min}$ if the smallest vertex cover of $G$ has size at most $A_{\min}$. As it turns out, $C_{\max}/C_{\min} = 1.0013$, which shows the NP-hardness of approximating the optimal $k$-means clustering cost within a factor of $1.0013$.

Given a set $G'$ of $N$ points in $\mathbb{R}^{3n}$, let $\mu$ be a uniform distribution on the $N$ points in $G'$ such that each point $p \in G'$ has weight $\frac{1}{N}$. Suppose we restrict the barycenter to have support $k$, where $k$ is the number of centers in the above $k$-means clustering instance. Then a set of $k$ centers $c_1, \ldots, c_k$ inducing clusters $C_1, \ldots, C_k$ on $G'$ that achieves cost $C$ for $k$-means clustering on $G'$ translates to a barycenter of support size $k$ that induces optimal transport cost $\frac{C}{N}$, where the weight of $c_i$ in the barycenter is $\frac{|C_i|}{N}$, for each $i \in [k]$.

Thus the optimal $k$-means clustering on $G'$ has cost $C$ if and only if the Wasserstein barycenter has cost $\frac{C}{N}$. Therefore, we immediately have the proof of Theorem 6.2.