# OpenReview forum: "Dimensionality Reduction for Wasserstein Barycenter"
_NeurIPS.cc/2021/Conference — NeurIPS 2021 Poster_

### Official Review · Reviewer_y9H3 · 2021-07-09

**Rating:** 7
**Confidence:** 3

**Summary:**

This paper introduces dimensionality reduction techniques and coresets for Wasserstein barycenters with fixed support size n. More precisely, motivated by the fact that exact algorithms have a complexity that depend exponentially on the dimension d, the authors first show that for p-Wasserstein barycenters, a random projection to a space of dimension $O(p^4/\epsilon^2 \log n / \epsilon)$ that preserves the cost of the barycenter within a $(1\pm \epsilon)$ factor can be obtained with high probability, independently of the original dimension d, the size of the distributions, and their number. Second, the authors show that for a given set of mesures $\mu_1, …, \mu_M$, there exists a subset of $\mathrm{poly}(d, n) /\epsilon^2$ size with corresponding weights such that for any target measure $\nu$, the original cost lies within $(1 \pm \epsilon)$ factor w.r.t. the weighted cost on this coreset.


**Limitations And Societal Impact:**

See main review.

No foreseeable societal impact.

**Main Review:**

Overall, I enjoyed reading this paper, which is clear and well-motivated. However, I have some questions and reservations regarding the section on coresets and the experiments, which I would like the authors to address.

### Strengths:

-  Given the exponential dependance of Wasserstein barycenter algorithms on the dimension, the paper is particularly well-motivated.
- The results on dimensionality reduction are quite strong. The fact that the target dimension only depends on n and $\epsilon$ (and  not e.g. the number of measures) is impressive.

### Weaknesses:

- It’s hard to understand how to build a coreset in practice without reading the appendix. Adding a more precise description of the algorithm in the paper could help in that regard. In particular, how much would that algorithm cost?  This would help understanding whether Theorem 5.2 is a theoretical result, or if can it be used in practice.
The experiments are not convincing:
- For the FACES experiments the choice $k=2$ seems to make it reduces the barycenter to the interpolation (at least for p=2). Besides, restricting to m=5 (FACES) and m=40 (MNIST) seems very strong, as in that case the solution is probably very different from the actual (unrestricted) barycenter.
- The coreset experiment is particularly problematic. For p=2 and Dirac measures, the Wasserstein barycenter is just the usual Euclidean mean, which makes the experiment a poor illustration of Theorem 5.2. It should be replaced with an experiment that actually involves the Wasserstein geometry.


### Other questions and comments:


- l.95-96 «  who showed that computing sparse Wasserstein barycenters… ».  As unconstrained Wasserstein barycenters are sparse, what is actually shown by Altschuler et al. is that computing Wasserstein barycenters is NP hard. As they even showed inapproxibility results, it seems that a fairer account of the contribution of the present paper (compared to Altschuler et al.) is rather that fixing the size of the barycenter does not alleviate the NP hardness of the problem.

- What would be the practical use of coresets? It seems that we need to have access to a Wasserstein barycenter to build one, hence contrary to the dimensionality reduction techniques they cannot be used to reduce the cost of computing barycenters

- Thm 5.2: it’s not very clear what is meant by p = O(1). Just that p is constant, or bounded, or not a function of n, d, k, etc. ?

- l.226-231: in the reconstruction phase, how do you handle the case where two points have the same projections?

- l.34-36: While true, this statement is a bit unfair towards the contribution of Altschuler et al. whose contribution was to give a poly(n, k) but exp(d) algorithm.


### Typos and minor comments

- Def 2.1: $w_1, …, w_n$ (not $w_j$)

- Def 2.1: « Solution » is a bit ambiguous because it does not necessarily correspond to a barycentre. Maybe something like « admissible pair » or « admissible solution » would be better?

- l. 150-153: why not use the notation $a_{ij}$ instead, to lift the ambiguity ?

- l.920-921 in the appendix there’s a typo (unfinished sentence)

- Some titles in the references are missing upper-case letters


**Time Spent Reviewing:**

4

---

> ### Author Response · Authors · 2021-08-09
> **Response to Reviewer y9H3**
>
> We thank you for your insightful comments and are glad that you enjoyed the paper. We note that we have made all of the minor corrections that you noted for the next version of the paper and address only the main questions here.
>
> > It’s hard to understand how to build a coreset in practice without reading the appendix.
>
> Due to space constraints, we made reference only to the general importance sampling procedure used for constructing coresets, as the implementation for the barycenter problem follows the general framework once the sensitivity bounds have been established. However, we agree that it would improve the clarity of the paper for readers unfamiliar with coresets for a brief review of the importance sampling procedure to be introduced, and we have added this to the main body in the next version of the paper. In the case this paper is accepted, we will also address this concern by further expanding the discussion on practical coreset construction in the additional content page that would be available in the camera-ready version.
>
>
> > Besides, restricting to $m=5$ (FACES) and $m=40$ (MNIST) seems very strong, as in that case the solution is probably very different from the actual (unrestricted) barycenter.
>
> We chose these parameter values for computational efficiency and simplicity of interpretation. Note that in the MNIST case, the barycenter can be interpreted as a distribution over ‘average’ digits since each of our input distributions represents one digit class. Similarly in the Faces dataset, our two distributions are left looking vs right looking faces so the barycenter distribution can be interpreted as a distribution over faces that ‘interpolate’ between left and right facing faces. This is a different use case than the typical one for barycenter experiments where each image is a distribution with point masses being the individual pixels. We chose our interpretation to get natural high dimensional datasets. Therefore in this setting, restricting the barycenter to a small support size is also meaningful. However, we envision that similar performance gains can be achieved if we use larger support sizes as suggested by our theoretical results.
>
> > The coreset experiment is particularly problematic. For $p=2$ and Dirac measures, the Wasserstein barycenter is just the usual Euclidean mean, which makes the experiment a poor illustration of Theorem 5.2. It should be replaced with an experiment that actually involves the Wasserstein geometry.
>
> Thank you for the feedback. The high level goal of our coreset experiments was to show that importance sampling can be much better than uniform sampling through some simple and illustrative examples. In practice, many real-world datasets represent "average-case" inputs that can be handled by uniform sampling, which has no provable guarantees. However, our coresets are constructed to handle adversarial inputs or "worst-case" inputs. Thus we chose to demonstrate our theoretical guarantees by constructing an adversarial dataset that typical uniform sampling approaches cannot handle.
>
> > What would be the practical use of coresets? It seems that we need to have access to a Wasserstein barycenter to build one, hence contrary to the dimensionality reduction techniques they cannot be used to reduce the cost of computing barycenters
>
> For the coreset construction, we need access to an $\alpha$-approximate solution to the barycenter problem, where $\alpha$ is an oversampling parameter that can even be super-constant, e.g., $\text{polylog}(n, d)$. To make use of this, we can construct a "rough" approximate solution on all of the distributions, then use this rough solution to construct a coreset with size independent of the number of original distributions. After this, we can refine our barycenter by performing computations with the coreset. This is useful when the initial number of distributions is large.
>
> > Thm 5.2: it’s not very clear what is meant by $p = O(1)$. Just that $p$ is constant, or bounded, or not a function of $n$, $d$, $k$, etc. ?
>
> The subset size includes a factor of $4^{p-1}$, so $p = O(1)$ should be thought of as saying that $p$ is a constant (and in particular, not a function of $n$, $d$, or $k$). In most applications, $p = 1$ or $2$ and in general, the choice of the loss function itself usually does not change with $n$, $d$, or $k$, so this is a reasonable assumption. See Theorem C.9 for the exact statement.
>
> > l.226-231: in the reconstruction phase, how do you handle the case where two points have the same projections?
>
> For the random JL projection maps that we use, this will occur with zero probability so we do not need to consider this issue. Specifically, conditioned on the guarantee of the JL Lemma, all distances are preserved up to a multiplicative factor of $1 \pm \epsilon$, which precludes two distinct points from being projected to the same point. If the random projection map does not satisfy the JL Lemma, then we cannot give guarantees.
>
> > l.34-36: While true, this statement is a bit unfair towards the contribution of Altschuler et al. whose contribution was to give a $\text{poly}(n, k)$ but $\exp(d)$ algorithm.
>
> We have added a remark that avoiding the curse of dimensionality is not the main contribution of their paper, which is instead to obtain a $\text{poly}(n, k)$ running time algorithm.
>
> > l.95-96 «  who showed that computing sparse Wasserstein barycenters… ». As unconstrained Wasserstein barycenters are sparse, what is actually shown by Altschuler et al. is that computing Wasserstein barycenters is NP hard. As they even showed inapproxibility results, it seems that a fairer account of the contribution of the present paper (compared to Altschuler et al.) is rather that fixing the size of the barycenter does not alleviate the NP hardness of the problem.
>
> Thank you, we have clarified this difference in the next version of the paper.

---

> > ### Comment · Reviewer_y9H3 · 2021-08-17
> > **Post-rebuttal comment**
> >
> > Thanks to the authors for answering my concerns.
> >
> > I have read all reviews and the answers from the authors.
> >
> > My opinion (which seems to be shared by other reviewers) is that the strong point of the paper are the theoretical results on dimensionality reduction, which are quite interesting and very relevant given the practical issues that are commonly encountered in OT applications. On the other hand, the presentation of the section on coresets could be improved.
> >
> > The experimental part is not as good as the theoretical part, but this is not necessarily a major issue for a theoretical paper. Thank you for further explaining the rationale of FACES and MNIST experiments. I am still quite unconvinced by the way the coresets experiment is presented. If the goal is indeed to illustrate the superiority of importance sampling compared to uniform sampling, then this should be clearly stated and the fact that the experiment is run in the Euclidean geometry on $\mathbb{R}^d$ should be emphasized. If the goal is to illustrate Wasserstein coresets, as is implicit in the way the experiments are currently presented, then the setting should be modified (the number of support points in each measure in particular) and the experiments re-run.
> >
> > I will maintain my score, and trust the authors to account for the issues raised by all reviewers.

---

### Official Review · Reviewer_GF2E · 2021-07-16

**Rating:** 7
**Confidence:** 3

**Summary:**

This paper proves that one can reduce the dimension of the Wasserstein barycenter problem for discrete measures, while ensuring an arbitrary small error for the loss function in the higher dimensional space $d$. More precisely, when the barycenter is restricted to a support with at most $n$ points, the measures $\mu_1,\ldots,\mu_k$ (with support of order $\mbox{poly}(n)$), can be mapped in a space of dimension $O(\log n)$ (independent of both $d$ and $k$) and still guaranteed a good solution.

**Limitations And Societal Impact:**

Limitations :
- The authors only provide bounds for the approximated barycenter when the probability distributions are mapped into a lower dimensional space through a Gaussian matrix. For more general JL projections, the results are only given in terms of cost functions.
- The construction of a JL projection should be better explained.

Minor comments :
- In the abstract, line 9, $k$ is not defined.
- line 1 : "geometric construct" should be "geometric construction".
- line 298 : $\omega(M)$.
- From my point of view, the term "partition" in Definition 2.1. is badly chosen since $(S_1,\ldots,S_n)$ does not form a partition of $\cup_{i=1}^k\mbox{supp}(\mu_i)$ since a point $x$ can belong to more than one $S_j$. If it was the case, mass splitting for the measures $\mu_i$ would not be allowed.
- In my opinion, the coresets experiments could be better presented. It is written that $x$ first takes the values $k= 50,000$ and then the values $x= 0, 1, 10$.
- As your results strongly rely on Johnson-Lindenstrauss Lemma, I recommend that you include it in the core of the paper.



**Main Review:**

Originality : To my knowledge, this paper is the first to propose a generalised method to reduce the complexity of computing Wasserstein barycenter. The idea is based on the reformulation of the discrete barycenter problem.

Quality : The theoretical analysis of the accuracy of the cost function (i.e. the function to be optimised in the Wasserstein barycentre problem) after projections of the probability measures onto a lower dimensional space is instructive and supports the claim that the dimensionality reduction through a JL projection can guarantee the correct order. These results also show that the dimension of the lower space does not depend on the initial dimension of the space, but only on the number of support points of the barycenter. Nevertheless, it could be of interest to validate the results of Theorem 3 for example, with respect to $n$ and $k$. Furthermore, it is not clear in practice how a JL projection is constructed.

Clarity : In my opinion, the paper is well written, but somehow the presentation of the results and the organisation of the paper could be improved, in particular Section 2 (some results are presented several time, with different names, see also "Minor comments" below). Finally, a concluded section might be added.

Significance : The proposed method can be used as a preprocessing step for any Wasserstein barycenter problem, and therefore is very relevant for applications in optimal transport. However, the performance of this method should be compared with dimensionality reduction algorithm for Wasserstein barycenter such as [Projection Robust Wasserstein Barycenters, Minhui Huang, Shiqian Ma, Lifeng Lai].


**Time Spent Reviewing:**

4

---

> ### Author Response · Authors · 2021-08-09
> **Response to Reviewer GF2E**
>
> Thank you for your feedback. We are especially pleased that you appreciated the novelty and significance of our work. We will add a roadmap detailing the organization of the paper to improve the presentation of the results in the next version of the paper. We have also incorporated the points raised in your minor comments section. We do believe there is a  significant confusion regarding the application of our results to various dimensionality reduction maps. Our results apply any map that satisfies the JL Lemma, not only random dense Gaussian matrices. In particular, we mention that sub-Gaussian matrices are valid random maps. We also describe the construction of a sparse matrix that permits a fast JL transformation in Appendix B.
>
> > Furthermore, it is not clear in practice how a JL projection is constructed.
>
> Random dense Gaussian matrices can be efficiently generated by generating each entry of the matrix independently at random from the (scaled) normal distribution. As mentioned above, we also describe the construction of a sparse matrix for fast JL projections in Appendix B.
>
> > The authors only provide bounds for the approximated barycenter when the probability distributions are mapped into a lower dimensional space through a Gaussian matrix. For more general JL projections, the results are only given in terms of cost functions.
>
> We remark in Line 193 that our results apply to *any* dimensionality reduction maps that satisfies the condition of the JL Lemma. This includes but is not limited to Gaussian matrices. For example, random sign matrices satisfy the condition of the JL Lemma and are easy to construct. Similarly, the sparse Hadamard matrix described for Corollary B.15 in Appendix B is a common approach that satisfies the condition of the JL Lemma while offering much better runtimes than dense Gaussian matrices. We will add descriptions of some of these representative examples in order to improve the clarity of the paper for readers not familiar with JL projections. We kindly ask you to rephrase this question if our response did not address your concerns.
>
> > the performance of this method should be compared with dimensionality reduction algorithm for Wasserstein barycenter such as [Projection Robust Wasserstein Barycenters, Minhui Huang, Shiqian Ma, Lifeng Lai].
>
> Thank you for pointing out this very relevant reference. As you noted, we propose a simple pre-processing tool to lower the ambient dimension of the problem. Since [Projection Robust Wasserstein Barycenters, Minhui Huang, Shiqian Ma, Lifeng Lai] proposes a coordinate descent method that has the potential to approximately solve the Wasserstein barycenter problem while avoiding an exponential dependency in the dimension, it can be combined with our pre-processing tool be even more effective, i.e., Line 42 and Line 205. Similarly, it can be combined with our coreset construction to lower the number of input distributions and thus have a lower runtime. We will add comparisons to this work in the next version of the paper.
>
> We thank you again for your comments. If you feel your main concerns have been addressed, we hope you will consider raising your score. Otherwise, we would also be happy to clarify any misconceptions we may have about your concerns.

---

> > ### Comment · Reviewer_GF2E · 2021-08-17
> > **Response to authors**
> >
> > Dear authors,
> >
> > Thanks for taking my comments into account and the clarification, I was under the impression that Theorem 4.3 (the one restricted to Gaussian matrices) was proposing a result on the optimal barycenter itself and not on the value of the objective cost, which caused the confusion.
> >
> > Also in my opinion, the addition of experimental details in the core of the document could improve the paper, as the proposed method is very promising.
> >
> > In view of the response, and as the authors have committed to clarify the presentation of their results, as well as adding comparisons in the experimental part, I have decided to increase my score.

---

### Official Review · Reviewer_SYrR · 2021-07-17

**Rating:** 7
**Confidence:** 3

**Summary:**

This paper present a method to reduce the dimension of k discrete measures defined on (at most) n support points in a real space of dimension d, by an O(log(n)) factor, while the optimum Wasserstein barycenter value maintains a bounded error from the optimal cost.

The main results of this work are given in Theorem 4.1 and 4.2. The paper presents the main plan of the technical proofs, but all the details of the proof are deferred to the online supplement of the paper. The two key ingredients of the proof are Johnson-Lindenstrauss (JL) projection results and the Kirszbraun theorem.

The paper contains also basic computational results on two standard datasets, the FACES and the MNIST datasets, which show the impact of the dimensionality reduction on the optimal value of the barycenter problem.


**Limitations And Societal Impact:**

It's ok.

**Main Review:**

The paper is very well written, and the recent literature on the topic is correctly reviewed. The organization of the paper gives first the basic notions to understand the contribution of the two main theorems given in Section 4.

The main complicating point in presenting the paper is to distinguish the case of “Wasserstein medians”, which are obtained when the underlying Wasserstein metric is computed for p=1, for the case p=2. Indeed, for p=2, all the computations are easier, and could also be explained by exploiting the interpolation properties of the Wasserstein-2 metric as done (heuristically) done in:

- Bouchet, P.Y., Gualandi, S. and Rousseau, L.M., 2020, September. Primal heuristics for Wasserstein Barycenters. In CPAIOR2020, pp. 239-255. Springer.

The authors clarify from the begging that their method is interesting when the number of support points “n” of the barycenter is significantly smaller than the space dimension of the support points (paragraph at lines 205-210). For instance, if we had to compute the barycenter of discrete measures supported in R^2 or R^3, the method would not provide any significant enhancements.

We were only confused in the computational results section. Honestly, I am used to seeing to images has discrete measure with support in R^2 (see for instance the very first paper on Wasserstein Barycenters [CD14]), while the authors use each image of the dataset as a point of R^d (with d=784 for MINST and d=4096 for images). Then what is “n”? the maximum number of “white” pixels over all the images? And what does represent the barycenter that you compute? The barycenter across the k classes? (k=2 for FACES and k=10 for MNIST? How did you compute the exact barycenter value? With Linear Programming or with a heuristic algorithm?

As a general question, what happens to your approach when the support of the barycenter is dense?

Finally, I have found the discussion about coresets (and the relative computational paragraph) unnecessary. While it surely adds value to the paper, it takes space for the more important results of Section 4, which could be used to give more details instead of presenting the plan of the proofs of the theorems, which remains the important contribution of this nice paper.

Minor comments:
-	Line 88: “dimension is needed”->”dimensions are needed” ?
-	Line 141: “a number OF heuristic…”
-	Line 156: Check the definition of \nu
-	Line 171 and 179: check the definition of \mu_iS
-	Theorem 4.3: what is \omega(M)? where is defined?
-	Check the spelling of “coreset” which is often typed as “corset” (eg. Line 305, 314,…)
-	Check all the references. Entries AC11a and AC11b are a duplicate. Check all the capitals, for instance, for Johnson-Lindenstrauss, Lipschitz, …
-	Figure 3 is Table and is not a Figure. The table should report also the standard deviations for the mean values.


**Time Spent Reviewing:**

10 hours

---

> ### Author Response · Authors · 2021-08-09
> **Response to Reviewer SYrR**
>
> Thank you for the positive assessment of our paper and for checking the paper thoroughly. We have integrated your suggestions in the minor comments section and performed another editorial pass checking for spelling/grammatical issues. We provide the following responses to your larger questions.
>
> > the authors use each image of the dataset as a point of $R^d$ (with $d=784$ for MINST and $d=4096$ for images). . Then what is “$n$”
>
> Here $n$ is the number of point masses in each distribution. For example in the MNIST experiments, each distribution is a digit class which is uniform over all the images in the class. Therefore, $n$ is roughly $10^3$ since we have that many images per class. This interpretation of image datasets is possibly more natural because it permits an easier way to encode the value of each pixel and hence perhaps why MNIST and FACES use this interpretation, though the tradeoff is that consequently, the distributions have high dimensions.
>
> > And what does represent the barycenter that you compute? The barycenter across the $k$ classes?
>
> In the MNIST case, we can interpret the barycenter as the distribution over ‘average’ digits  since each of our input distributions are one digit class. Similarly in the Faces dataset, our two distributions are left looking vs right looking faces so the barycenter distribution is a distribution over faces that ‘interpolate’ between left and right facing faces.
>
> > How did you compute the exact barycenter value? With Linear Programming or with a heuristic algorithm?
>
> We use the implementation of Ye et al. as stated in Section 6. Their algorithm uses a variant of gradient descent to compute the barycenters. Note however that as stated in Section 2 and Algorithm 1, our method for dimensionality reduction works with **any**algorithm for barycenter computation since we show that the cost of any solution, as defined in Definition 2.1, is preserved arbitrarily well. Therefore, we believe that similar results can be achieved with other implementations. We chose the Ye et al. since it was very accessible, well documented, and had been used in prior experimental papers.
>
> > As a general question, what happens to your approach when the support of the barycenter is dense?
>
> When the support of the barycenter is dense, e.g., $n$ is exponential in the ambient dimension $d$, the main runtime bottleneck is not incurred by the “curse of dimensionality” and thus the main focus should not be on techniques for reducing the dimension of $d$. We believe this choice of parameters is an interesting regime, but perhaps orthogonal to our line of work.

---

> > ### Comment · Reviewer_SYrR · 2021-08-18
> > **Response to authors**
> >
> > Dear authors,
> > thanks for your detailed replies.
> >
> > I read all the other reviews and all of your answers, and, overall, I confirm my score (in short: nice theoretical paper, but experiments could/should be improved).
> >
> > Best regards.

---

> > ### Comment · Reviewer_SYrR · 2021-08-24
> > **Acknowledge the author's response**
> >
> > We thank the authors for their answers to all the reviews, and we confirm our score.

---

### Official Review · Reviewer_1Zyc · 2021-07-17

**Rating:** 6
**Confidence:** 4

**Summary:**

This work tackles the Wasserstein barycenter computation problem. It 1) shows how using a randomized projection can alleviate the curse of dimensionality for barycenter computation, and 2) provides a way to use coresets to reduce the number of input distributions to compute the barycenter of.

**Limitations And Societal Impact:**

I looked, but did not find much of a discussion on the limitation of this work. I do not see potential negative social impacts.

**Main Review:**

The paper builds upon the recently-developed theory by MMR19, originally devised for k-means, to apply dimensionality reduction to the barycenter problem that is better than naively applying JL. In particular the reduced dimension is independent of the number of input distributions.

This work also includes results on the NP-hardness of the barycenter problem, a tight lower bound given by JL for the related OT distance computation problem, as well as using coresets to reduce the number of input distributions to compute the barycenter of.

My main concern is the very limited number of experiments supporting the theory, based on both the code provided and the experiments described. Only two small datasets are used, FACES and MNIST, the former of which only contains 698 images, and both are from the image domain. And the runtime on the reduced-dimension MNIST barycenter computation is ~three quarters (73%) of the original runtime, with no variance across measurements given (if multiple measurements were done for timing). Also only one implementation of barycenter computation was tested.

The empirical evidence on coresets construction is similarly paltry, with experiments on only one synthetic dataset and limited experimental settings.

**Time Spent Reviewing:**

10

---

> ### Author Response · Authors · 2021-08-09
> **Response to Reviewer 1Zyc**
>
> Thank you for your comments. We are glad to hear that you appreciated the theoretical strength of our work. We understand your concerns with experiments beyond MNIST and FACES. However, we remark that our experiments serve as an empirical proof-of-concept to demonstrate the validity of our main theoretical statements. We did not attempt to optimize the running time of any algorithms used and it is possible that other datasets beyond MNIST and FACES can exhibit even better improvements. In particular, we note that since our dimensionality reduction technique serves as a pre-processing tool to improve the performance of any end-to-end algorithm, it may be beyond the scope of our work to perform a comprehensive evaluation of which practical algorithms are particularly optimized for lower input dimensions.
>
> > with no variance across measurements given (if multiple measurements were done for timing).
>
> Yes, we performed multiple measurements to evaluate the average and central tendencies. In general, the variance of the time measurements were at least an order of magnitude smaller than the average time taken. We will explicitly report these statistics in the next version of the paper.
>
> > Also only one implementation of barycenter computation was tested.
>
> As stated in Section 2 and Algorithm 1, our method for dimensionality reduction works with *any* algorithm for barycenter computation since we show that the cost of any solution, as defined in Definition 2.1, is preserved arbitrarily well. Therefore, we believe that similar performance gains can be achieved with other implementations. We chose the Ye et al. since it was very accessible, well documented, and had been used in prior experimental papers.
>
> > The empirical evidence on coresets construction is similarly paltry,
>
> Thank you for your feedback. The high level goal of our coreset experiments was to show that importance sampling can be much better than uniform sampling through some simple and illustrative examples. In practice, many real-world datasets represent "average-case" inputs that can be handled by uniform sampling, which has no provable guarantees. However, our coresets are constructed to handle adversarial inputs or "worst-case" inputs. Thus we chose to demonstrate our theoretical guarantees by constructing an adversarial dataset that typical uniform sampling approaches cannot handle.
>
> We thank you again for the clarifying comments on the empirical section of the paper. We will add greater details (e.g. variance across measurements) as well as graphics for the coreset experiments to the next version of the paper, and we hope that the logic behind our choice of experiments is now more clear as our work is primarily theoretical. We hope these alleviates your concerns; if not, we would be happy to engage in a more thorough discussion. Otherwise, we hope that you will consider raising your score if there are no further concerns.

---

> > ### Comment · Reviewer_1Zyc · 2021-08-19
> > **Thank you for your response**
> >
> > Thank you, authors, for your response. I have read your responses, as well as other reviews. I understand that the focus of the paper is on theory, but given that the demonstrated empirical improvement is only shown on two small datasets under one domain, and the improvement not particularly large, as noted in the original review, I still believe that more experimentation is warranted to provide convincing corroboration of the theory. This can be done without involving a comprehensive evaluation of which algorithms are optimized for lower input dimensions. As such, I believe that the paper is above the acceptance threshold for its theoretical strengths, but should be improved by more thorough experimentation.

---

### Author Response · Authors · 2021-08-10
**Thanks to all reviewers**

We thank the reviewers for their thoughtful comments and valuable feedback. We especially appreciate the positive remarks, such as
* The paper is very well written (Reviewer SYrR)
* The proposed method can be used as a preprocessing step for any Wasserstein barycenter problem, and therefore is very relevant for applications in optimal transport (Reviewer GF2E)
* In my opinion, the paper is well written (Reviewer GF2E)
* Overall, I enjoyed reading this paper, which is clear and well-motivated (Reviewer y9H3)
* The results on dimensionality reduction are quite strong. The fact that the target dimension only depends on $n$ and
(and not e.g. the number of measures) is impressive (Reviewer y9H3)
* Given the exponential dependance of Wasserstein barycenter algorithms on the dimension, the paper is particularly well-motivated. (Reviewer y9H3)

We provide our responses to the specific questions of each reviewer below. We hope our answers resolve all initial questions and concerns raised by the reviewers and we will be most happy to answer any remaining questions!

---

### Decision · Program_Chairs · 2021-09-28

**Decision:**

Accept (Poster)

**Comment:**

All the reviewers mentioned that it is an interesting paper, which is clearly written, with simple to understand yet non trivial theoretical results. The numerical simulations are a bit weak, but this being put aside, all the reviewers are supportive of acceptance. As a side note, a paper which is related to the proposed method and is worth discussing is “B. Muzellec, MC, Subspace Detours: Building Transport Plans that are Optimal on Subspace Projections, Neurips 2019.”

**Consistency Experiment:**

NeurIPS has a long history of experimentation. In 2014, NeurIPS ran an experiment in which 10% of submissions were reviewed by two independent committees to quantify the randomness in the review process. This year, we repeated a variant of this experiment to see how the quality of the review process has changed over time.  This paper was part of the experiment and was therefore assigned to two committees (consisting of reviewers, an Area Chair, and a Senior Area Chair) that reached independent decisions.  If both committees made the same recommendation, this recommendation was followed. If a single committee recommended acceptance, the paper was accepted (with the exception of a few cases in which the other committee identified what we considered a fatal flaw, e.g., an error in a key result).

This copy’s committee reached the following decision: **Accept (Poster)**

The other committee assigned to the paper recommended **Reject**.  You can find the other set of reviews, along with any follow up discussion with the authors here:
https://openreview.net/forum?id=d4Lo6PhbKA